# The bacterial RNA polymerase-associated CarD protein couples promoter activity to DNA supercoiling

David Forrest ✉, Emily A. Warman & David C. Grainger ✉

RNA polymerase (RNAP) must unwind duplex DNA prior to transcription initiation. In bacteria, unwinding starts at a DNA motif called the promoter −10 element. Specifically, non-template strand bases interact with RNAP to trigger the process. A protein called CarD can support −10 element opening in many microbes. Whilst this most often activates transcription, CarD may repress if the DNA open complex is too stable. For *Rhodobacter sphaeroides*, a purple photoheterotrophic alpha-proteobacterium, CarD is particularly important as many −10 elements have a key DNA sequence defect. Here, we use genomic tools to map transcription initiation, binding of RNAP, CarD interactions, and DNA topology, globally in *R. sphaeroides*. We show an association between CarD-controlled transcription, and negative DNA supercoiling, which alters global gene expression if perturbed. Using biochemical tools, we show that promoter co-regulation by CarD and supercoiling results from the −10 element defect inherent to *R. sphaeroides* promoters. If this flaw is corrected, regulation by CarD and supercoiling is disrupted. As supercoiling dissipates during stress, we suggest CarD couples housekeeping transcription to the environment via DNA topology.

Throughout life, transcription is catalysed by a highly conserved enzyme called RNA polymerase (RNAP). In bacteria, the core enzyme has 5 subunits: an α dimer, β, β′ and ω[1]. Whilst this core is catalytically active, accurate recognition of promoter DNA sequences requires a dissociable σ factor. Frequently, bacteria encode multiple σ factors, with different promoter specificities. Most transcription is directed by the primary housekeeping σ factor, designated σ[70] in *Escherichia coli*, which recognises two hexameric DNA motifs: the −35 and −10 elements. These sequences, named according to their position upstream of the transcription start site (TSS, +1) have a consensus sequence of 5′-TTGACA-3′ and 5′-TATAAT-3′, respectively. Proteins in the σ[70] family share 4 structural domains: σ1, σ2, σ3, and σ4[2]. Domain σ4 recognises the −35 element, whilst σ2 contacts the −10 sequence[3–5]. Upon binding the −10 motif, σ2 facilitates DNA melting and an open complex is formed. Specifically, bases −11A and −7T flip out of the DNA base stack and occupy pockets within σ2[4,6,7]. Hence, positions −11A and −7T are highly conserved. The general

principles of promoter recognition and DNA unwinding are common to many bacteria.

*Rhodobacter sphaeroides* is a purple photoheterotrophic alpha-proteobacterium and has a housekeeping σ factor called σ[93][8]. Surprisingly, over 50% of *R. sphaeroides* −10 elements have a sequence defect and lack the −7T base[9]. Despite the implications for DNA opening, such promoters drive high-level expression of essential genes, including rRNA operons. This is permitted by CarD, a transcription factor discovered in *Mycobacterium tuberculosis*[10]. Found in diverse bacterial phyla, CarD stabilises DNA opening and so compensates for the missing −7T base[11–13]. To do this, CarD binds the β subunit and uses a conserved tryptophan as a wedge to help separate the DNA strands[12–14]. Whilst this typically activates transcription, over-stabilisation of DNA opening can be inhibitory[11]. For example, CarD represses its own transcription in this way[15].

Separation of DNA strands can also be influenced by nucleic acid topology[16]. For instance, over- or under-twisting DNA generates

School of Biosciences, University of Birmingham, Edgbaston, Birmingham, UK. ✉e-mail: d.a.forrest@bham.ac.uk; d.grainger@bham.ac.uk

positive or negative supercoils respectively[17]. Strand unwinding is easier in the latter case. Inside cells, biological processes can generate DNA supercoils. Thus, during transcription, RNAP creates negative and positive supercoils, behind and ahead of the transcription elongation complex respectively[18,19]. This is primarily due to a phenomenon known as the twin supercoiling domain model[19]. Briefly, the crowded intracellular environment prevents rotation of the bulky RNAP molecule around the double helix during enzyme translocation. To compensate, the DNA twists and supercoils arise. Globally, correct DNA organisation is maintained by topoisomerases. For example, DNA gyrase introduces negative DNA supercoils at sites across the chromosome. Even so, global supercoiling levels can change dramatically in response to environmental conditions[20–24].

In this work, we have mapped global patterns of supercoiling and transcription in *R. sphaeroides*. We identify exact sites of transcription initiation, the presence of $\sigma^{93}$, and the binding of CarD. The data reveal frequent synergy between CarD-dependent gene control and DNA supercoiling. For example, if DNA gyrase is inhibited, CarD is released from many promoters, rendering them less active. This can be recapitulated in vitro, with purified CarD and RNAP. In this way, we show that co-dependence of promoters, on CarD and negative DNA supercoiling, results from the −10 element defects (i.e., the absent −7T base) outlined above. Hence, correcting the motif circumvents the simultaneous need for both CarD and under-twisted DNA. Similarly, our data show promoters, co-repressed by CarD and negative supercoiling, also have defective −10 elements. If this flaw is remedied and the DNA relaxed, CarD can revert to its more frequent role of transcription activation. We show that genes activated by CarD and negative supercoiling are frequently linked to housekeeping functions, including protein synthesis. We suggest CarD links the expression of these genes to environmental conditions, via negative supercoiling, which dissipates during slow growth.

## Results

### Global analysis of transcription initiation in Rhodobacter sphaeroides

Global patterns of transcription in *R. sphaeroides* are poorly defined, and current genome-scale approaches have hardly been used. To address this, we mapped global CarD and $\sigma^{93}$ binding using ChIP-seq. Simultaneously, we measured total transcription with RNA-seq and identified transcription start sites with cappable-seq. The data, collected during aerobic growth, are integrated as genome-wide maps in Fig. 1a. The results are further summarised in Supplementary Data 1 and 2. We identified 8,150 TSSs, including 1248 of the 2252 sites previously identified by Myers et al. (Supplementary Fig. 1)[25]. The greater number of TSSs identified here likely represents the improved sensitivity of cappable-seq compared to older methods[26]. Overall, the distribution of TSSs in *R. sphaeroides* resembles other proteobacteria, including *E. coli* and *Salmonella* Typhimurium[27–29]. Thus, 61% of TSSs are located within coding DNA (Fig. 1b). Of intergenic TSSs, 89% are correctly orientated to drive mRNA transcription. The remaining intergenic TSSs were incorrectly orientated or between convergent genes (Fig. 1c, left). We note that recent work has suggested transcription may sometimes initiate independently of a canonical promoter in vitro[30,31]. However, available evidence indicates this is rare in vivo, as multiple processes remove the post-termination RNAP complex from the DNA[31,32].

### Bidirectional promoters occur frequently in Rhodobacter sphaeroides

A subset of TSSs were closely spaced and on opposite DNA strands (supplementary data 3). These correspond to bidirectional promoters, with inherent −10 element symmetry, from which transcription can proceed in either possible direction. Such promoters are found throughout prokaryotes, but have an unusual configuration in *R. sphaeroides*[27]. Briefly, for bidirectional promoters in most bacteria, −10

element positions −11A and −7T co-locate to the same base pair, on opposite DNA strands. As such, TSSs are separated by 18 bp. In *R. sphaeroides*, such divergent TSSs are most often 23 bp apart. In this configuration, promoter positions −12T and −11A reciprocally co-locate (Supplementary Fig. 2a, b). The likely reason is the frequent absence of base −7T in *R. sphaeroides*. Overall, 8.8% of *R. sphaeroides* promoters are bidirectional. Similar to other bacteria, bidirectional promoters are much more likely to be intergenic and between divergent genes than their directional counterparts (Fig. 1c, right)[27,28].

### Promoters associated with CarD have sub-optimal −10 elements

To understand $\sigma$ factor and CarD association with each TSS, we examined our ChIP-seq data. In total, we identified 1167 and 588 peaks for $\sigma^{93}$ and CarD, respectively. Of the 8150 TSS found by cappable-seq, 39% were bound by $\sigma^{93}$, and 48% of these sites were also associated with CarD (Fig. 1d, e). Only 2% of CarD-bound TSSs were not also recognised by $\sigma^{93}$. It is notable that intergenic TSSs are much more likely to be associated with $\sigma^{93}$ or CarD (Fig. 1b).

To better understand the properties of promoters bound by CarD, we made DNA sequence logos (Fig. 1f)[33]. Note that, prior to alignment, promoter regions were split into discrete sections. This avoids sequences being out of register, due to variation in spacer and discriminator length, during alignment. In subsequent logos, the height of each letter indicates sequence preference and information content. Compared to all other promoters, those associated with CarD have defective −10 elements due to poor conservation of base − 7 T. Conversely, the − 12T and − 11A bases, also critical for −10 element function, were conserved regardless of CarD binding. Given the role of base −7T in DNA unwinding, it seems likely that promoters associated with CarD have inherent defects for strand separation. This is consistent with prior reports that CarD controls transcription by stabilising the open complex[12–14].

### Promoters associated with CarD exhibit increased negative DNA supercoiling

We reasoned that negative DNA supercoiling might impact CarD-dependent control of transcription. To understand this, we used psora-seq[34,35]. Briefly, psoralen is a molecule that preferentially intercalates with underwound DNA. Hence, in combination with next-generation sequencing, psoralen was used to map global patterns of DNA supercoiling (Supplementary Data 4). Figure 2a shows maps of psoralen binding for each *R. sphaeroides* chromosome. The maps are presented at both high resolution (track 2, signal averaged across 5 kb bins) and low resolution (track 3, 100 kb bins). At low resolution, chromosome 1 partitions into origin and terminal halves, with the latter being more negatively supercoiled. The smaller chromosome 2 has no large-scale supercoiled domains. Rather, the entirety of chromosome 2 is more negatively supercoiled than chromosome 1. For both chromosomes, high-resolution maps reveal local patterns of supercoiling. In part, these can be explained by transcription. For example, *R. sphaeroides* has 3 rRNA operons; *rrnA* is located near the origin of chromosome 1, whilst *rrnB* and *rrnC* occupy a similar position on chromosome 2. The highest psoralen binding values locate to these highly transcribed regions (labelled in Fig. 2a). Negative supercoiling was also associated with transcribed DNA more generally (compare psora-seq and RNA-seq data in Fig. 2a). To understand the relationship between CarD binding and supercoiling, we compared the psora-seq and ChIP-seq profiles (Fig. 2b). CarD-bound promoters are significantly enriched for psoralen binding and so negative supercoiling, compared to unbound and random genomic loci. Some examples are shown in Fig. 2c. In each case, the CarD-associated TSS is within a negatively supercoiled DNA region. To confirm this association was specific to CarD, and not a property of all promoters, we also determined psoralen binding values for $\sigma^{93}$ peaks, with or without an overlapping signal for CarD. Promoters bound by both CarD and $\sigma^{93}$ were significantly

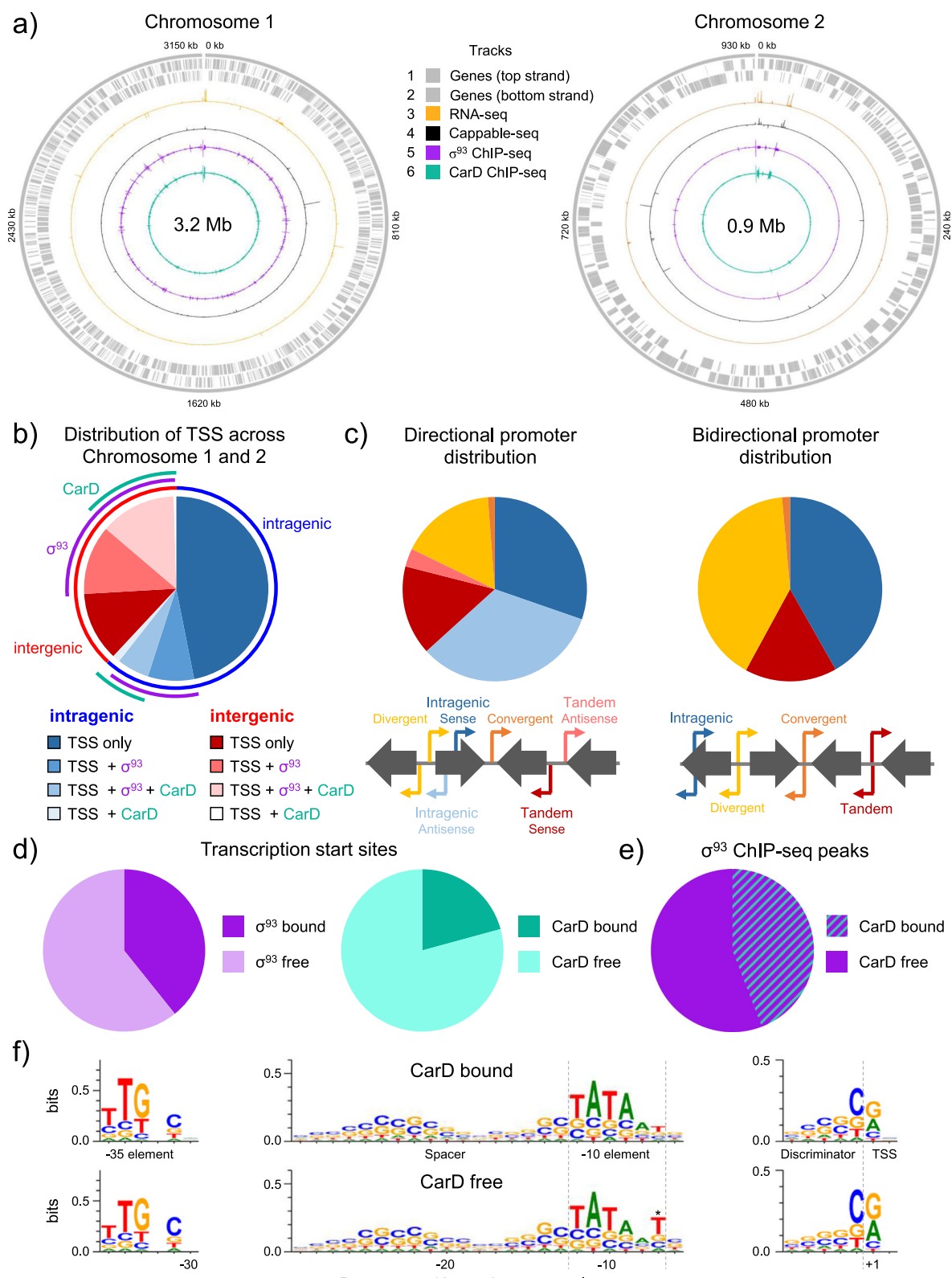

more negatively supercoiled than those bound by σ⁹³ alone (Supplementary Fig. 3).

**Decreased DNA supercoiling impacts CarD occupancy at transcription initiation sites**

Our data show that CarD-associated promoters are enriched for negative DNA supercoiling (Fig. 2b). To investigate the relevance of

this, we repeated our ChIP-seq, RNA-seq, and psora-seq assays, with subinhibitory concentrations of novobiocin, a competitive inhibitor of DNA gyrase. We expected novobiocin to reduce negative supercoiling and so the psora-seq signal[20,36]. We also expected changes in DNA supercoiling to impact patterns of CarD binding and transcription. A summary of the psora-seq data, for different groups of chromosomal loci, is shown in Fig. 3a. As expected, treatment with novobiocin

**Fig. 1 | Global maps of transcription initiation and gene expression in *R. sphaeroides*. a** The panel schematics show *R. sphaeroides* chromosomes I and II. Genes are shown as grey lines on the top or bottom DNA strands (outer two tracks). Plotted alongside are data from RNA-seq (orange), cappable-seq (black), and ChIP-seq (purple or teal) experiments. For tracks 3 to 6, positive and negative values indicate sequencing read depth for the top and bottom DNA strands. **b** Protein composition and location of transcription start sites. The pie chart indicates the distribution of transcription start sites (TSSs) between genes and non-coding sequences. For each of these categories, TSSs are further categorised according to their association with $\sigma^{93}$ and/or CarD. Source data are provided as a Source Data file. **c** Genomic context of directional (left) and bidirectional (right) promoters. Pie chart segments are coloured according to the schematics below. These indicate promoter (bent arrow) positions with respect to genes (block arrows). Source data

are provided as a Source Data file. **d** Coincidence of all transcription start sites with $\sigma^{93}$ or CarD ChIP-seq peaks. Source data are provided as a Source Data file. **e** Coincidence of all $\sigma^{93}$ and CarD ChIP-seq peaks. Source data are provided as a Source Data file. **f** Sequence properties of promoters differ according to CarD binding. The TSS and discriminator were aligned according to TSS position. Both the −10 and −35 elements were aligned based on the sequence best matching the consensus, whilst also allowing for a small degree of variability in position (see materials and methods). Letters indicate bases, according to normal conventions, and are sized according to information content (bits). Labels on each x-axis indicate positions relative to the TSS ( +1, bottom) or key promoter elements (top). The asterisk highlights base −7T that is usually absent from CarD-bound promoters. Since they are more likely to impact gene expression, sequence logos were generated from promoters within 300 bp of a gene start codon.

---

caused significant DNA relaxation across all subsets of chromosomal regions: the average psora-seq signal for CarD-bound, CarD-unbound, and random loci significantly decreased by 26%, 50% and 38%, respectively (Fig. 3a). To understand the impact on CarD binding, we compared ChIP-seq signals for each CarD-bound promoter, with and without novobiocin. The results are shown as a volcano plot in Fig. 3b, where each data point represents a single ChIP-seq peak. There is a clear trend for novobiocin to reduce CarD binding. Most likely, this is because promoter open complexes are destabilised, leading to the release of CarD and RNAP.

### Reduced DNA supercoiling predominantly inhibits transcription at CarD-bound promoters

Since both CarD and negative supercoiling enhance strand separation, we expected novobiocin to alter $\sigma^{93}$ binding at, and transcription from, many CarD-bound promoters. For example, in situations where RNAP cannot open the duplex efficiently, and requires assistance from both CarD and negative supercoiling, novobiocin should inhibit transcription. Conversely, if CarD and negative supercoiling over-stabilise DNA unwinding, transcription could increase. To test these predictions, we quantified novobiocin induced changes in $\sigma^{93}$ binding (Fig. 3c) and transcription (Fig. 3d) at CarD targeted genes. Significant reductions in $\sigma^{93}$ binding, and transcription, were detected at 138 and 260 CarD gene targets, respectively. Less frequently, $\sigma^{93}$ binding (24 peaks) and transcription (108 genes) increased. Hence, negative supercoiling more often activates, rather than represses, transcription from CarD-bound promoters. Two example regions are shown in Fig. 3e. In both cases, novobiocin induces DNA relaxation, loss of both CarD and $\sigma^{93}$ binding, and lower gene expression.

To understand the relationship between altered CarD binding and transcription, the data in Fig. 3b were reannotated. This was done according to gene expression changes adjacent to each CarD peak. The reannotated data are shown in Supplementary Fig. 4. Data point position and colour indicate changes to CarD binding, and adjacent gene expression, respectively. There is a clear tendency for reduced CarD binding and transcription inhibition to coincide (Supplementary Fig. 4a, blue data points). Conversely, increased transcription is associated with either no change, or a modest increase, in CarD binding (red data points). We conclude that there are two major classes of CarD-associated promoter. The first and largest corresponds to the 260 genes where transcription decreases upon DNA relaxation. We refer to these as CarD-bound relaxation-repressed promoters. The second class controls the 108 genes where transcription increases upon DNA relaxation. We refer to these as CarD-bound relaxation-activated promoters.

### Different classes of CarD-bound promoter have similar sequence properties

To understand the properties of the two promoter classes, we generated DNA sequence logos (Fig. 3f). Visual comparison revealed no

major difference between the motifs. Thus, we used a simple position weight matrix (Supplementary Fig. 4b) to score core promoter elements and determine the significance of any subtle variation. We also quantified GC-content of the discriminator region, located between the −10 element and TSS, that also influences DNA unwinding[37]. Overall, the two promoter classes were remarkably similar (Fig. 3g). Of the small differences detected, statistical significance was only apparent for the −10 element score. We conclude that relaxation-repressed and relaxation-activated promoters, bound by CarD, cannot be easily identified by gross sequence properties. Rather, CarD-bound promoters, as a whole, are notable for their shared −10 element defect; absence of base − 7T (Fig. 1f). Hence, we speculate that DNA supercoiling, rather than DNA sequence alone, could also be a key consideration. Indeed, we note that relaxation-repressed promoters bound by CarD tend to be in regions of lower negative supercoiling compared to their relaxation-activated counterparts (Supplementary Fig. 4c).

### Activation of transcription by CarD in vitro requires negative DNA supercoiling

To more precisely understand the role of DNA supercoiling, we first focused on the most common class of CarD-bound promoter; those repressed by DNA relaxation. The *rrnB*P1 promoter is an excellent example (Supplementary data 5) having previously been identified as CarD activated[9,10,38]. To understand the role of supercoiling, we used in vitro transcription assays. The template for transcription was plasmid pRLG770, having the *R. sphaeroides rrnB*P1 promoter upstream of the *E. coli rrnB* T1 and T2 terminators[9]. Before use, plasmids were treated with topoisomerase I, to remove all DNA supercoiling. These DNA molecules have low mobility during electrophoresis (Fig. 4a, lane 1). We then repeated topoisomerase treatment in the presence of ethidium bromide. In this scenario, negative supercoils are introduced in a manner dependent on the concentration of the DNA intercalator (Supplementary Fig. 5a)[39]. Hence, by using different amounts of ethidium bromide, we generated a set of plasmids with increasing amounts of negative supercoiling (Fig. 4a, lanes 2-6). For comparison, raw negatively supercoiled plasmid DNA, isolated directly from cells, is shown in lane 7. Note that standard gel electrophoresis cannot separate plasmid topoisomers with high, but different, levels of supercoiling (lanes 4-6). However, the expected differences are revealed in the presence of chloroquine (Supplementary Fig. 5b). These differences are indicated schematically in Fig. 4a. Results of in vitro transcription assays, with the different templates, are shown in Fig. 4b. The RNAI transcript, generated from the plasmid replication origin, serves as an internal control. In the absence of CarD, both *rrnB*P1 and RNAI transcription increased in response to greater negative supercoiling (Fig. 4b, lanes 1−7). However, this increase was modest for *rrnB*P1 (compare *rrnB* and RNAI transcript levels in lanes 1−7). Addition of CarD had no impact on RNAI transcription but specifically increased *rrnB* promoter activity (compare lanes 1−7 with 8−14). Furthermore,

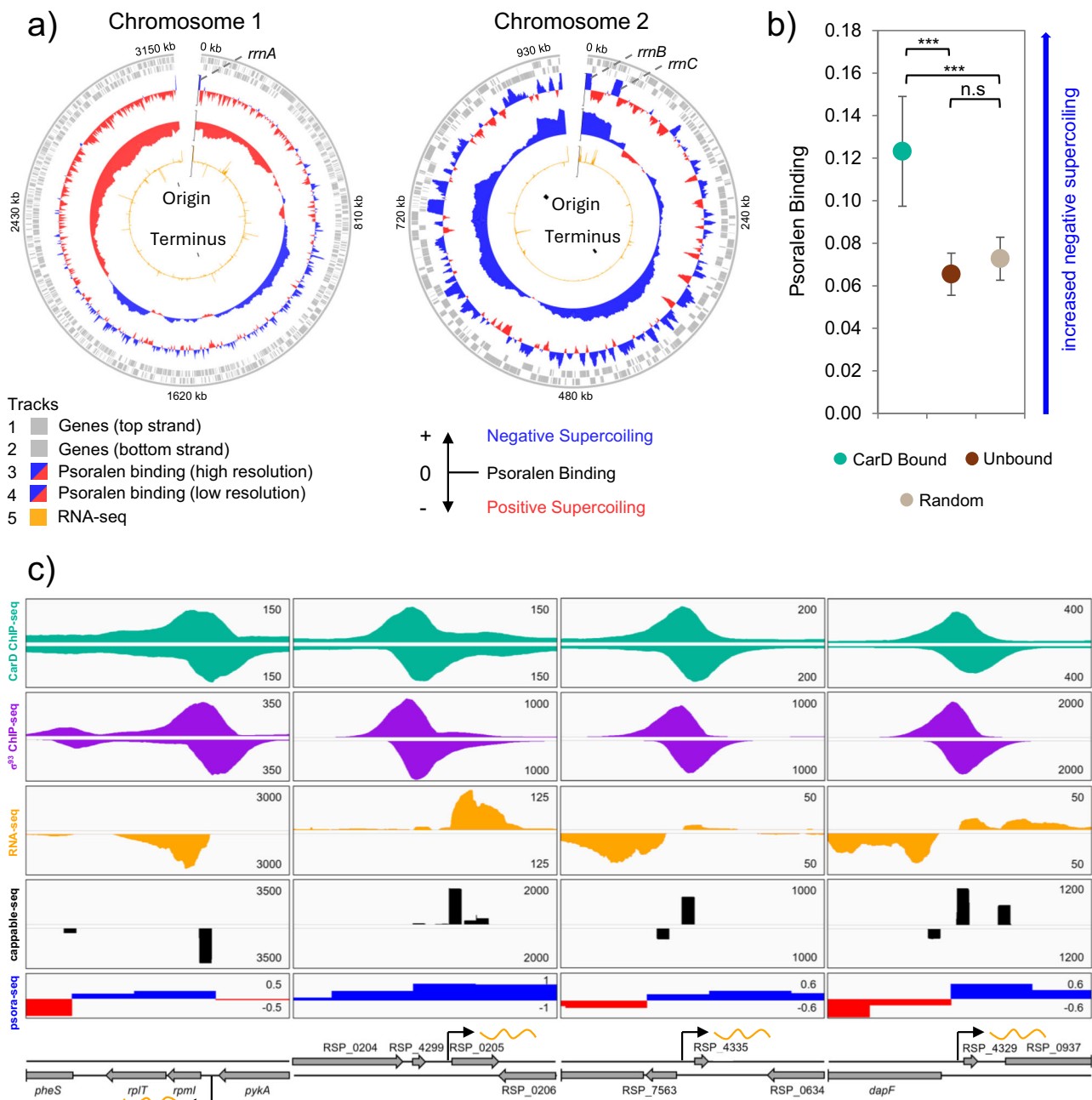

**Fig. 2 | CarD-bound promoters are in regions of elevated negative supercoiling.**
**a** A global map of DNA supercoiling in *R. sphaeroides*. The panel schematics show *R. sphaeroides* chromosomes I and II. Genes are shown as grey lines on the top or bottom DNA strands (tracks 1 and 2). Plotted alongside, are data from psora-seq (tracks 3 and 4) or RNA-seq (track 5) experiments. For the latter, positive and negative values indicate sequencing read depth for the top and bottom DNA strands. Psoralen binding values, derived from the psora-seq data, indicate negative (blue) or positive (red) supercoiling. Note that the same psora-seq data are presented at both high (5 kb moving average) and low (100 kb moving average) resolution. High-resolution psoralen binding values were capped at 0.77 to −0.77 for chromosome 1 and 0.36 to -0.36 for chromosome 2. This allows better visualisation of DNA supercoiling at non-ribosomal operons. Similarly, the displayed RNA-seq read depth is restricted to 4634 to −4634 for chromosome 1 and 1774 to −1774 for chromosome 2. Predicted locations of origin (Ori) and terminal (Ter) regions are depicted on the inner-most track. **b** CarD-bound genomic regions have

increased negative DNA supercoiling. Mean psoralen binding values were derived from two biological replicates. An interval plot showing mean psoralen binding values for the indicated chromosomal regions. Note that only genomic regions containing a TSS were used. Error bars show the 95% confidence interval. The significance of differences between psoralen binding signals, for all chromosomal regions in each category, was determined using a homoscedastic two-tailed *t* test. The *P*-values for CarD bound DNA, compared to unbound or random DNA, are 7.31499e⁻⁶ and 0.0001 respectively. *P*-values of < 0.001 are marked ***. Source data are provided as a Source Data file. **c** Examples of negative supercoiling at CarD-bound promoters. Compiling ChIP-seq, RNA-seq, cappable-seq and psora-seq demonstrates negative supercoiling at CarD-bound promoters. Graphs show normalised sequencing read depths for top and bottom DNA strands, with maximum read depths for each strand displayed. Genes are shown as block arrows, and are labelled at the bottom of each panel. Bent arrows indicate the position of the CarD-associated promoter, and wavy lines indicate RNA transcripts produced.

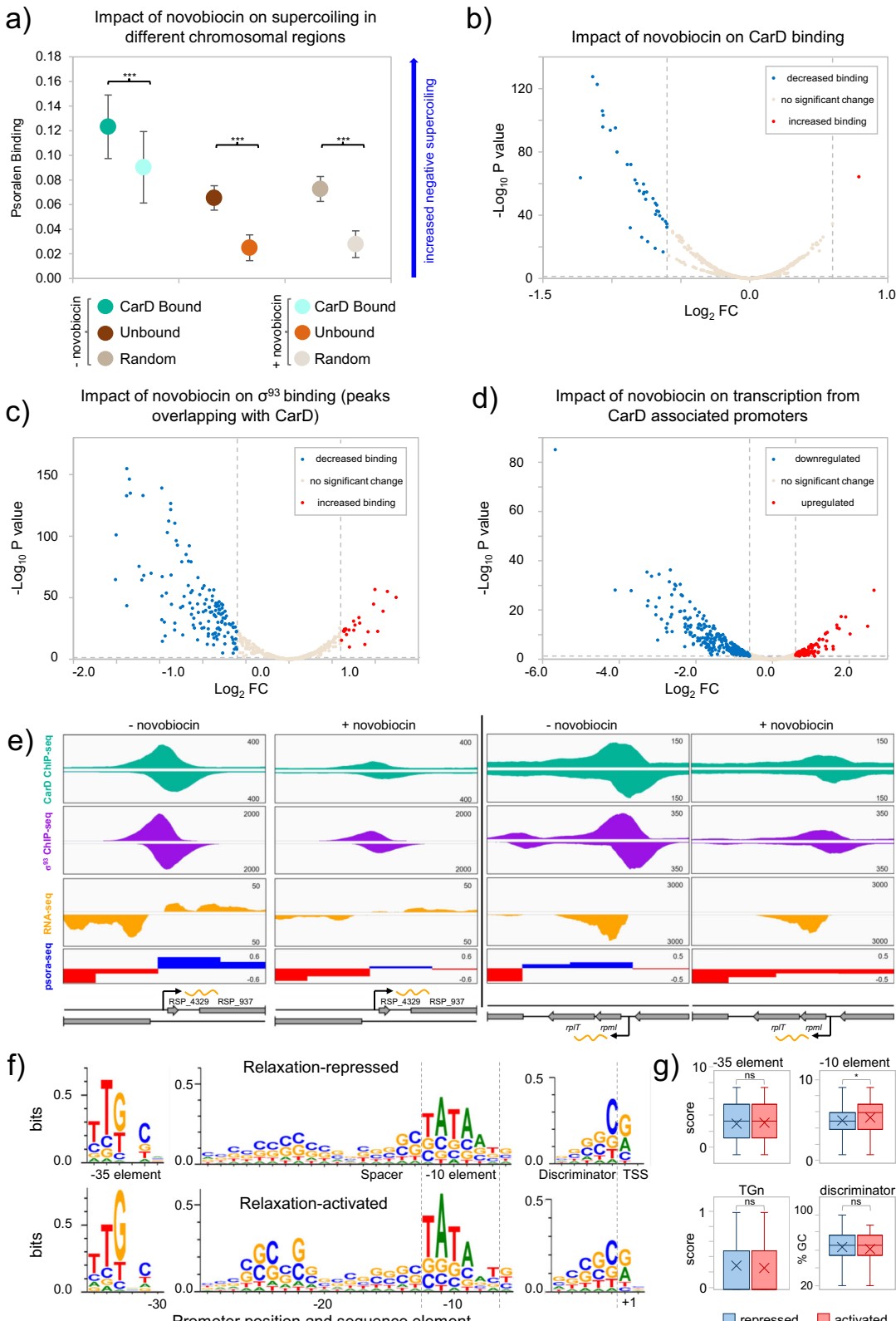

CarD was only able to act on DNA that was negatively supercoiled to a sufficient level. Thus, CarD had little or no impact on transcription from the relaxed (lane 8) or modestly supercoiled (lanes 9 and 10) DNA templates. Conversely, CarD activated *rrnB* transcription by >8-fold when negative supercoiling was sufficiently high (lanes 11-13). Critically, the level of negative supercoiling in raw plasmid DNA, isolated directly from cells, was sufficient for near maximal activation by CarD

(lane 14). A quantification of *rrnB* transcript levels, in the presence and absence of CarD, is shown in Fig. 4c. In related experiments, similar results were obtained with the *rpsL* promoter, P*rpsL*, that drives expression of a small ribosomal protein subunit (Supplementary Fig. 6, lanes 9–16). Thus, CarD could activate P*rpsL* in the presence (lanes 13–16) but not absence (lanes 9–12) of negative supercoiling. Consistent with our genome-scale data, we conclude that activation by

**Fig. 3 | Relaxation of supercoiling alters CarD binding and promoter activity in vivo. a** Novobiocin relaxes DNA supercoiling. Exponentially growing *R. sphaeroides* were treated with 2 µg ml⁻¹ novobiocin before psora-seq. Interval plots show mean psoralen binding values for CarD-bound, CarD-unbound, and random genomic regions with and without novobiocin. Mean psoralen binding values were derived from two biological replicates. Only genomic regions with TSSs were included. Error bars show the 95% confidence interval. The significance of differences between psoralen binding signals for chromosomal regions in each category, with or without novobiocin treatment, was determined using a paired two-tailed *t* test. CarD bound, unbound and random P-values are $4.59e^{-5}$, $2.13e^{-32}$, and $3.17e^{-37}$, respectively. *P*-values of < 0.001 are marked ***. Source data are provided as a Source Data file. **b** Novobiocin induced DNA relaxation reduces CarD binding at most promoters. Fold changes in ChIP-seq read depth, at CarD binding sites, with and without novobiocin, are shown as a volcano plot. Each point represents a single CarD binding site. Increased and reduced CarD binding is shown in red and blue, respectively. The threshold for significance was a $\log_2$ fold change (FC) of 0.6 (i.e., a 1.5-fold change) and a corrected *P*-value of < 0.05. Source data are provided as a Source Data file. **c** Novobiocin induced DNA relaxation reduces σ⁹³ binding at most CarD-associated promoters. As for Fig. 3b, but each point represents σ⁹³ binding at a site also bound by CarD. Source data are provided as a Source Data file. **d** Novobiocin induced DNA relaxation reduces transcription at most CarD-associated promoters. The panels each show a differential gene expression analysis comparing novobiocin treated to normally grown cells. Each data point represents

a gene expressed from a CarD-bound promoter. Source data are provided as a Source Data file. **e** Novobiocin induced DNA relaxation reduces CarD and σ⁹³ binding to downregulate gene expression. The panel shows example regions of the *R. sphaeroides* genome, with or without treatment with 2 µg ml⁻¹ novobiocin. CarD ChIP-seq, σ⁹³ ChIP-seq, RNA-seq, and psora-seq data are shown as graphs. The graphs plot normalised sequencing read depths for the top and bottom DNA strands, with the maximum read depth for each strand displayed. Increased or decreased psoralen binding reflects negative or positive supercoiling, depicted in blue and red, respectively. Genes are depicted as grey arrows and labelled. Data for genes RSP_4329 (left) and *rpmI* (right) are shown. **f** Sequence properties of CarD-bound promoters subject to repression (top) and activation (bottom) by novobiocin treatment. **g** CarD-bound promoters repressed by novobiocin have better matches to the −10 element consensus sequence. The box plots compare scores for the indicated promoter elements, for all novobiocin-repressed and novobiocin-activated promoters, also bound by CarD, where a higher score indicates a better match to the consensus. For the discriminator region, GC-content, rather than score, is shown. The promoters scored are derived from the TSS determine from two biological replicates. The cross shows the mean value, whilst the horizontal line indicates the median. The box indicates the 25th–75th percentile. The whiskers indicate minimum and maximum values. The significance of differences between promoter scores was determined using a homoscedastic two-tailed *t* test. −10 element repressed − activated *P*-value was 0.048. *P*-values of < 0.05 are marked *. Source data are provided as a Source Data file.

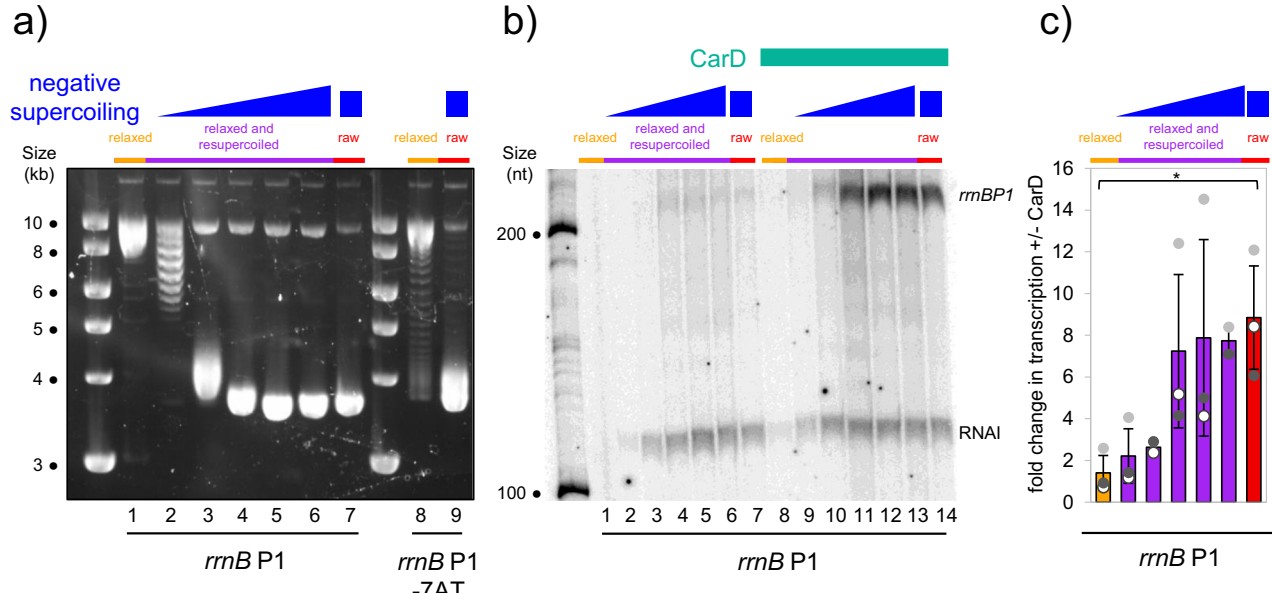

**Fig. 4 | CarD requires negative DNA supercoiling to activate transcription in vitro. a** Differential supercoiling of DNA templates for in vitro transcription. The CarD-dependent ribosomal promoter *rrnB* P1, and its mutant derivative -7AT, were cloned in plasmid pRLG770. Prior to using these plasmids as templates for in vitro transcription, they were treated with different combinations of topoisomerase I and ethidium bromide to manipulate levels of DNA supercoiling. Crude levels of template supercoiling were determined by agarose gel electrophoresis. Briefly, DNA relaxed with topoisomerase I has a low mobility during electrophoresis (lanes 1 and 8). However, if DNA is relaxed in the presence of ethidium bromide, subsequent removal of the DNA intercalator induces DNA supercoiling. This occurs in a concentration dependent manner, and lanes 2–6 correspond to plasmids treated with 2.5, 7.5, 12.5, 25 or 50 µM ethidium bromide. Lane 7 shows raw and unrelaxed midiprep DNA isolated from bacterial cells. Approximate supercoiling levels are shown above the gel image. Completely relaxed, relaxed and resupercoiled, as well as raw midiprep DNA, are also labelled. See Supplementary Fig. 5 for further details and direct comparison to sample analysis using chloroquine-gel electrophoresis.

Uncropped gel images are provided as Source Data. **b** Activation of transcription by CarD requires negative DNA supercoiling. The figure shows results of in vitro transcription assays using the *R. sphaeroides* RNAP σ⁹³ holoenzyme. Transcription is driven from the *rrnB*P1 promoter cloned in plasmid pRLG770. Crucially, the DNA template used for each individual reaction was differently supercoiled. Briefly, both lanes 1–7, and lanes 8–14, used the templates shown in lanes 1–7 of panel (**a**), respectively. The *rrnB*P1 and RNAI (internal control) transcripts are indicated. **c** Quantification of in vitro *rrnB*P1 promoter activity relative to levels of DNA supercoiling. The amount of *rrnB*P1 transcript obtained, using each template, was quantified with and without CarD. Values are mean fold changes with error bars showing standard deviation. Individual data points from replicates one (light grey), two (open), and three (dark grey) are shown as dots. Note that second lowest and highest ethidium bromide concentrations were omitted from replicates one and two, respectively. Calculated with a two tailed, homoscedastic student's *T* test, the *P*-value comparing relaxed to raw DNA is 0.016. *P*-values of < 0.05 are marked *. Source data are provided as a Source Data file.

CarD frequently requires sufficient negative supercoiling. The most likely explanation is that this DNA topology favours DNA unwinding and allows CarD to overcome the energetic barriers imposed when base −7T is absent.

### An improved rrnBP1 sequence responds differently to CarD and DNA supercoiling

We reasoned that *rrnB*P1 co-dependence on CarD and negative supercoiling might be removed by introducing the preferred −7T base. For example, CarD may activate such an optimised promoter when the DNA is relaxed. For comparison, we also examined the impact of a deleterious −12A mutation. Usually thymine, this base is transiently unpaired during DNA unwinding. Contact with the template strand is then restored prior to a stable open complex being established[6,40]. The various *rrnB*P1 derivatives tested are shown in the top half of Fig. 5a, and their differently supercoiled derivatives are in Supplementary Fig. 5c. Results of in vitro transcription assays are in Fig. 5b and data from three replicate experiments are quantified in Fig. 5c, which shows fold changes in transcription with and without CarD. As expected, wild-type *rrnB*P1 was only active in the presence of CarD and negative supercoiling (Fig. 5b, lanes 1–4). The -12A mutation rendered the promoter inactive in all conditions (lanes 5–8). Strikingly, the −7T mutation allowed full activation by CarD with a relaxed DNA template (compare lanes 4 and 10, also see Fig. 5c). Overall promoter activity increased substantially in the presence of negative supercoiling (lane 11) and CarD was unable to stimulate substantial further transcription (lane 12, and Fig. 5c). The impact of the −7T mutation was abolished when combined with the -12A base change (lanes 13–16, and Fig. 5c). We conclude that the requirement of CarD for negative supercoiling can be overcome by the −7T mutation. Similarly, CarD can override the absence of −7T if DNA has the correct topology. However, promoters lacking -12T cannot be activated by CarD.

### Repression of transcription by CarD in vitro requires negative DNA supercoiling

We next turned our attention to the less common class of CarD-bound promoter; those activated by DNA relaxation. Our genome-scale data predict that CarD and negative DNA supercoiling should synergistically reduce transcription at such promoters (Fig. 3 and Supplementary Fig. 4). To test this, we again used in vitro transcription assays. The template for transcription was plasmid pRLG770, having the *R. sphaeroides carD* promoter, P*carD*, upstream of the *E. coli rrnB* T1 and T2 terminators[9]. Note that P*carD* remains bound by CarD, upon novobiocin treatment, and transcription increases (Supplementary Data 5). Conversely, CarD was previously shown to repress P*carD* in the context of a negatively supercoiled DNA template in vitro[15]. Taken together, and consistent with our earlier speculation, these data suggest CarD may switch roles, from repressor to activator of P*carD*, upon DNA relaxation. The results of in vitro transcription assays are shown in Fig. 5b lanes 17-20. Consistent with our prediction and our genome-scale data, P*carD* was activated by CarD when the DNA template was first relaxed using topoisomerase I (lanes 17 and 18). Conversely, CarD repressed transcription, from the same promoter, in the context of a negatively supercoiled template (lanes 19 and 20). The likely explanation is that negative supercoiling and CarD, together over-stabilise P*carD* opening whilst, in the absence of supercoiling, sub-optimal opening creates CarD dependence. If correct, promoter mutations, that enhance DNA opening, may negate P*carD* activation in the context of a relaxed DNA template.

### An improved PcarD sequence responds differently to CarD and DNA supercoiling

To test our prediction, we repeated our in vitro transcription analysis using P*carD* having the -7 T mutation. For comparison, we again tested

the -12A mutation. The P*carD* derivatives tested are shown in Fig. 5a (lower half). The results are shown in Fig. 5b (lanes 21–32), and quantification is in Fig. 5d. The -12A mutation abolished promoter activity in all conditions (Fig. 5b, lanes 21-24). The -7T mutation, expected to stabilise DNA opening, rendered P*carD* moderately active in the absence of CarD and negative DNA supercoiling (lane 25). As predicted, this transcription was not further activated by CarD (lane 26 and Fig. 5d). In the presence of negative supercoiling, overall promoter activity increased substantially (lane 27) but remained subject to CarD-mediated repression (lane 28 and Fig. 5d). Thus, P*carD* co-repression, by CarD and negative supercoiling, is not negated by the -7T mutation in the same way as *rrnB*P1 activation is disrupted. We speculate P*carD* lacks base -7T to permit nuanced transcriptional changes in response to diverse DNA topologies.

### Chimeric promoters derived from rrnBP1 and PcarD are CarD activated

The ability of CarD and negative supercoiling to co-activate transcription at some promoters, whilst co-repressing others, is curious. There is no marked sequence difference between the promoter classes (Fig. 3f, e). Furthermore, our experiments with *rrnB*P1 and P*carD* indicate DNA topology, rather than sequence, may determine the regulatory impact of CarD. To test this further, we made a series of chimeric promoters. Hence, *rrnB*P1 sequence elements were replaced with equivalent P*carD* motifs. We also examined the impact of the -7T mutation and DNA topology in this context. The chimeric promoter sequences are shown in Fig. 6a and results of in vitro transcription assays, using these DNA templates, are in Fig. 6b. Quantification of data from replicate experiments, showing fold changes in transcription, with or without CarD, are in Fig. 6c. Consistent with prior experiments, activation of *rrnB*P1 required both CarD and negative supercoiling (lanes 1–4) and this requirement was removed by the -7 T mutation (lanes 5–8). Introduction of the P*carD* 5′-TGn-3′ motif, which facilitates RNAP binding but not DNA opening, rendered *rrnB*P1 more active. However, activation by CarD still required DNA supercoiling (lanes 9–12 and Fig. 6c). The P*carD* discriminator, which is more AT-rich than the equivalent *rrnB*P1 sequence, should facilitate DNA opening[37]. Consistent with this, CarD was able to significantly activate the promoter in the absence of negative supercoiling (lanes 13–14 and Fig. 6c). However, both supercoiling and CarD were required for full activation (lanes 15-16). Together, the P*carD* extended -10 hexamer, and more AT-rich discriminator, allowed substantial *rrnB*P1 activation by CarD in the absence of negative supercoiling (lanes 17-18 and Fig. 6c). Indeed, CarD could only further activate transcription modestly when negative supercoils were present (lanes 19-20 and Fig. 6c). When further combined with the −7T mutation, we detected high basal *rrnB*P1 activity (lane 21). This was further activated by CarD (lane 22) but only in the absence of negative supercoiling (lanes 22–24). It is notable that no changes render *rrnB*P1 subject to repression by CarD. We conclude that DNA topology may impact the regulatory role of CarD more than promoter sequence alone.

### CarD-bound promoters control key housekeeping functions

Genes encoding factors for information storage and protein synthesis are often regulated in response to growth rate[41,42]. Further, in starved cells, levels of negative DNA supercoiling are lower[22,43,44]. Hence, CarD might link the expression of such genes to growth rate via supercoiling changes. To test this, we quantified gene functions linked to reduced CarD binding at promoters upon DNA relaxation (blue points in Fig. 3b). As a control, we also identified gene functions associated with σ[93]-bound promoters lacking CarD. The gene functions are categorised in Fig. 7. Genes expressed from promoters, where CarD binding is reduced by DNA relaxation, are enriched for functions related to

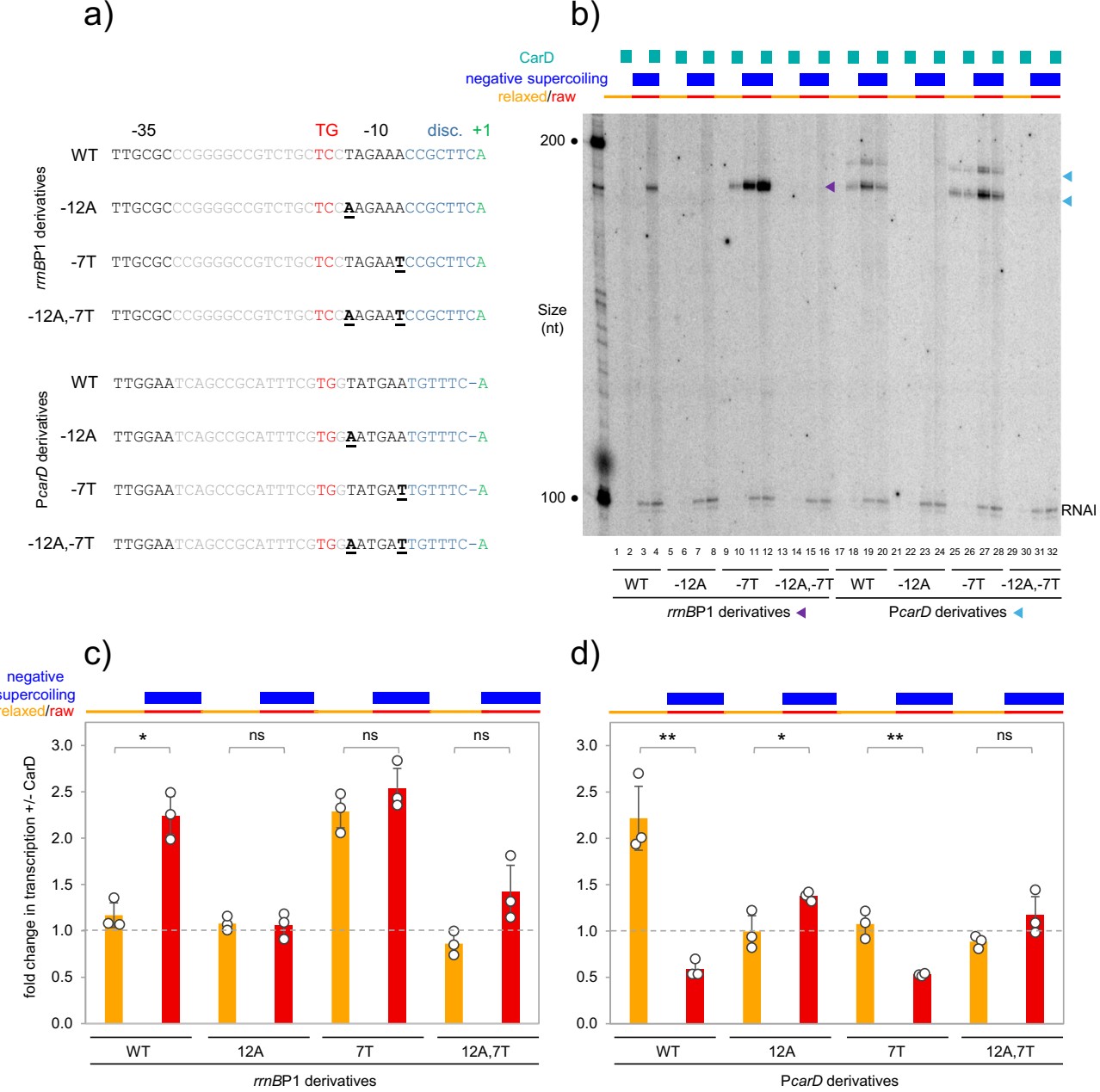

**Fig. 5 | The -7T mutation impacts promoter co-dependence on CarD and DNA supercoiling for both activation and repression of transcription. a** Sequences of the *rrnB*P1 and P*carD* promoters. Different promoter elements are labelled, and mutations are underlined. **b** The figure shows results of in vitro transcription assays using the *R. sphaeroides* RNAP σ[93] holoenzyme. Transcription is driven from the indicated derivatives of *rrnB*P1, or P*carD*, cloned in plasmid pRLG770. The DNA templates used were either pre-relaxed, using topoisomerase I, or negatively supercoiled raw templates isolated from cells (Supplementary Fig. 5c). Note that transcripts of two different lengths are observed for experiments with P*carD*. Since production of both is abolished by the −12A mutation, these must originate from the same promoter. Most likely, partial read-through of the downstream T1 terminator occurs before elongation finally ends at terminator T2. Uncropped gel images are provided as Source Data. **c**, **d** The bar charts show quantification of fold changes in transcription, induced by CarD, for either the relaxed or negatively supercoiled (raw) version of each DNA template. The dashed horizontal line indicates no change. Bars above and below this line represent activation and repression of transcription, respectively. Values are mean fold changes from three independent replicates, with error bars showing standard deviation. Dots show data from individual replicates. The significance of differences between fold changes was determined using a homoscedastic two-tailed *T* test. For wild-type *rrnB*P1, the relaxed compared to raw DNA P-value is 0.0035. For wild type P*carD*, the 12 A derivative, and the 7 T variant, equivalent respective *P*-values are 0.0029, 0.037 and 0.0033. *P*-values of < 0.05 or < 0.01 are marked * and **, respectively. Source data are provided as a Source Data file.

information storage and processing. Specifically, functions related to the synthesis of ribosomes and tRNAs (categories a and d, Fig. 7) are overrepresented. Note that, whilst other gene categories are linked to CarD-bound promoters and similarly impacted by DNA supercoiling, their functional categories are not specifically enriched (compare Fig. 7a, b).

## Physiologically relevant changes in DNA supercoiling modulate gene control by CarD

To test our model further, we first examined CarD binding to, and transcription from, 3 relaxation-repressed promoters, identified in our genomic assays, in living cells. Two of the associated genes belong to the enriched functional categories: RSP_4322 and *rmpI*, which are

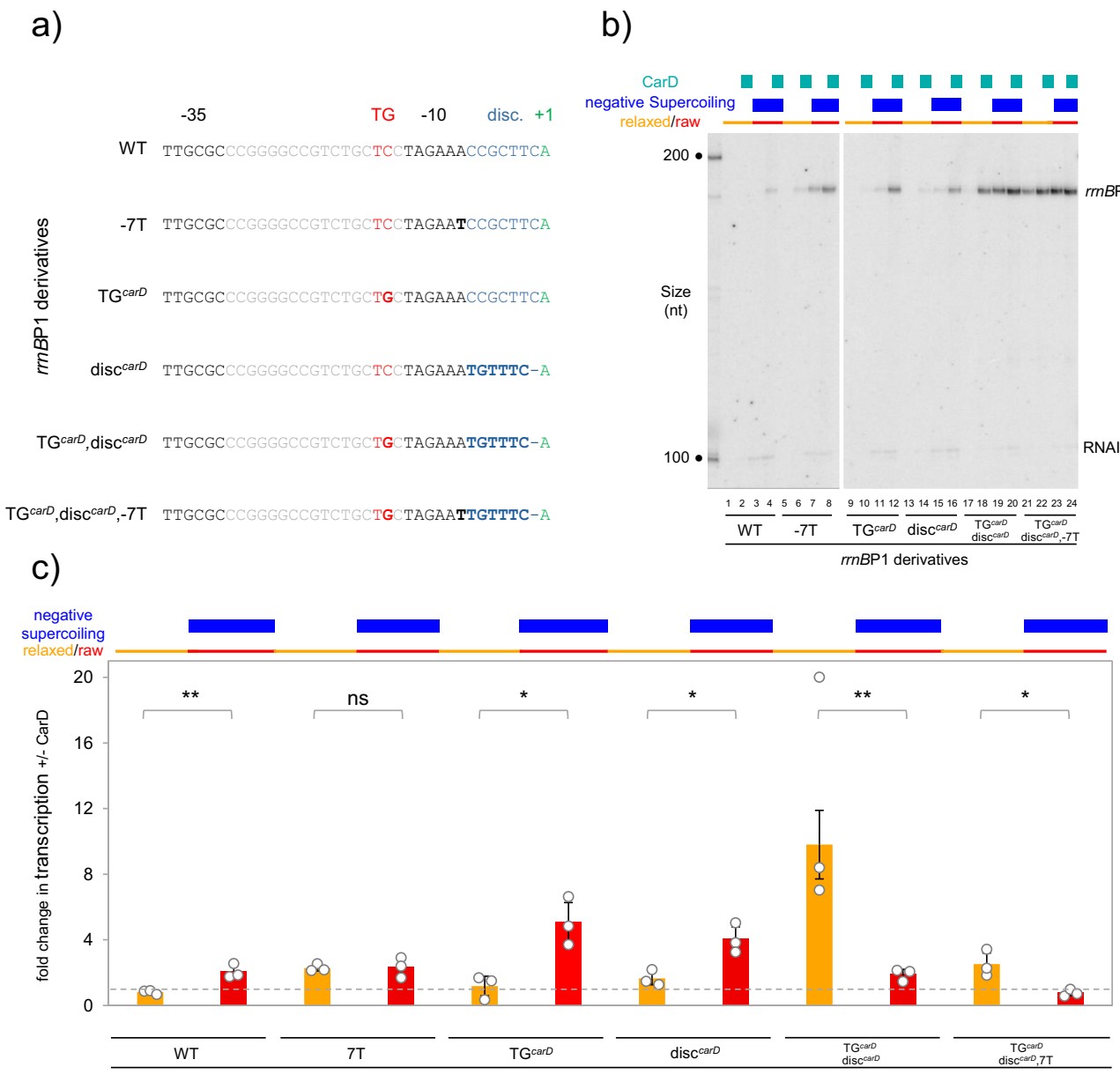

**Fig. 6 | Chimeric *rrnB*P1 derivatives respond differently to CarD and DNA supercoiling. a** Sequence of *rrnB*P1 and chimeric derivatives. Different promoter elements are labelled, and mutations are underlined. In chimeric derivatives, the indicated region of P*carD* was used to replace the equivalent section of *rrnB*P1. The impact of the −7T mutation was also investigated. **b** The figure shows results of in vitro transcription assays using the *R. sphaeroides* RNAP σ[93] holoenzyme. Transcription is driven from the indicated promoter cloned in plasmid pRLG770. The DNA templates used were either pre-relaxed, using topoisomerase I, or negatively supercoiled raw templates isolated from cells. Uncropped gel images are provided as Source Data. **c** The bar charts show quantification of fold changes in transcription,

induced by CarD, for either the relaxed or negatively supercoiled (raw) version of each DNA template. The dashed horizontal line indicates no change. Bars above and below this line represent activation and repression of transcription, respectively. Values are mean fold changes from three independent replicates, with error bars showing standard deviation. Dots show data from individual replicates. The significance of differences between fold changes, comparing relaxed and raw DNA, was determined using a homoscedastic two-tailed *T* test. The respective *P*-values for wild type *rrnB*P1, TG*carD*, disc*carD*, TG*carD*disc*carD* and TG*carD*disc*carD*7T are 0.0091, 0.015, 0.016, 0.0061 and 0.023. *P*-values of < 0.05 or < 0.01 are marked * and **, respectively. Source data are provided as a Source Data file.

implicated in tRNA biogenesis and ribosome function, respectively. Gene RSP_1952 encodes a cold shock protein, is transcribed from a CarD-bound promoter, and is also repressed upon DNA relaxation. Results comparing signals from ChIP-qPCR (to measure CarD binding) and RT-qPCR (to infer mRNA levels) assays, for growing and starved cells, are shown in Supplementary Fig. 7a, b. As expected, levels of CarD binding and transcription are dramatically reduced in starved cells when supercoiling decreases. However, these data must be interpreted with caution; levels of CarD and σ[93] are both lower in

starved cells, and this must also contribute (Supplementary Fig. 7c, d). Hence, to further validate our model, we took an additional approach. Briefly, *R. sphaeroides* is capable of aerobic or photosynthetic growth[45]. The latter case, cell division is much slower, and this likely impacts supercoiling. To test this, we used plasmid pBBRBB to transform *R. sphaeroides*. Transformants were cultured, either aerobically or photosynthetically, before plasmid DNA was extracted. Levels of supercoiling were then determined using chloroquine gel electrophoresis. As expected, greater negative supercoiling was apparent during faster

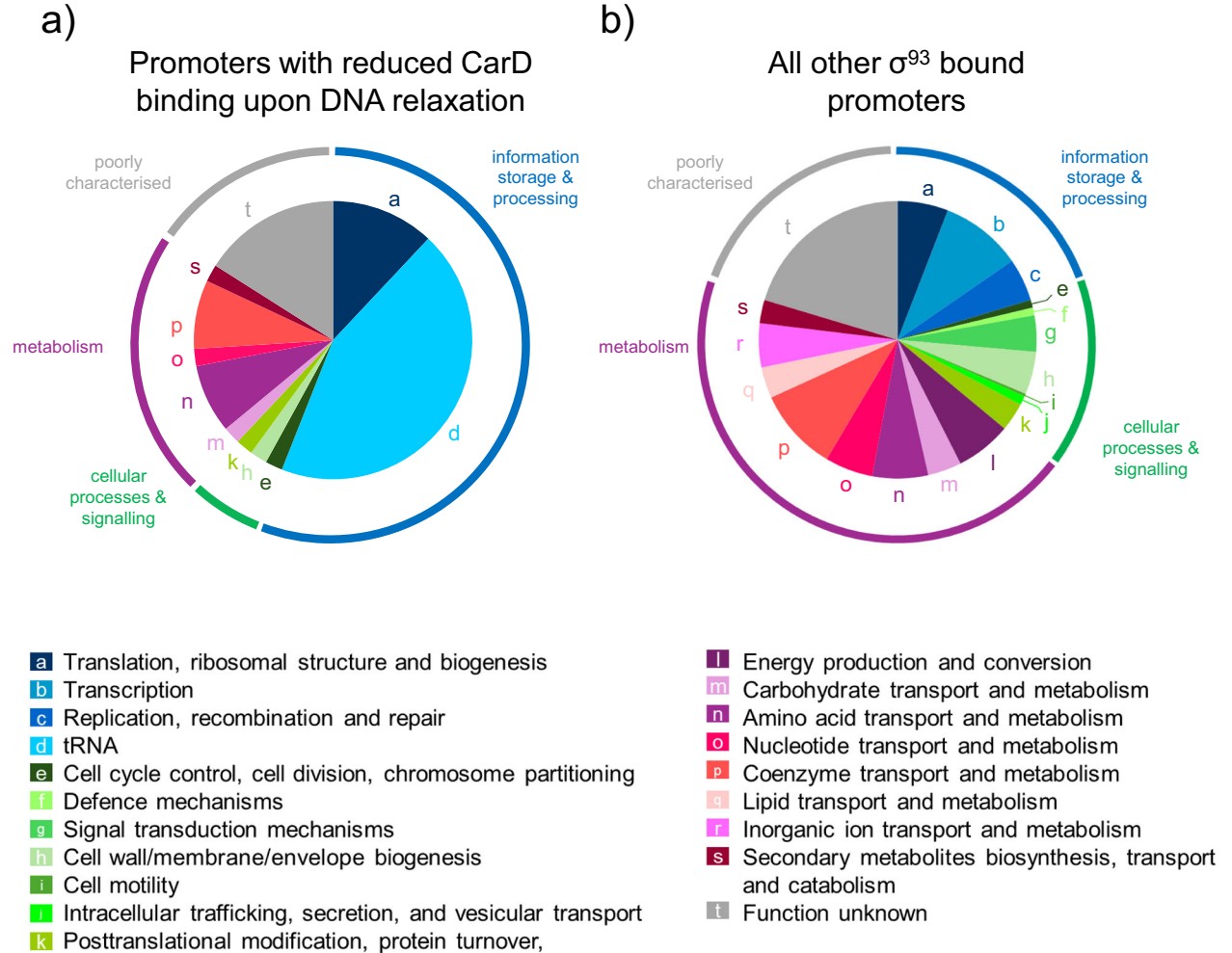

a) Promoters with reduced CarD binding upon DNA relaxation

b) All other σ⁹³ bound promoters

**a** Translation, ribosomal structure and biogenesis
**b** Transcription
**c** Replication, recombination and repair
**d** tRNA
**e** Cell cycle control, cell division, chromosome partitioning
**f** Defence mechanisms
**g** Signal transduction mechanisms
**h** Cell wall/membrane/envelope biogenesis
**i** Cell motility
**j** Intracellular trafficking, secretion, and vesicular transport
**k** Posttranslational modification, protein turnover, chaperones

**l** Energy production and conversion
**m** Carbohydrate transport and metabolism
**n** Amino acid transport and metabolism
**o** Nucleotide transport and metabolism
**p** Coenzyme transport and metabolism
**q** Lipid transport and metabolism
**r** Inorganic ion transport and metabolism
**s** Secondary metabolites biosynthesis, transport and catabolism
**t** Function unknown

**Fig. 7 | CarD and negative supercoiling activate genes for ribosome biogenesis and translation. a** Genes targeted by CarD and σ⁹³, and exhibiting reduced CarD binding upon novobiocin treatment, were categorised based on associated KEGG terms. Source data are provided as a Source Data file. **b** An equivalent analysis for genes targeted by σ⁹³ but not CarD. Source data are provided as a Source Data file.

aerobic growth (Supplementary Fig. 7e). To understand if these physiologically relevant changes in supercoiling could impact transcription activated by CarD, we used the extracted plasmids as templates for in vitro transcription. Our logic was that pBBRBB contains numerous housekeeping promoters, which are frequently bound by CarD and repressed by DNA relaxation. Furthermore, this avoids the confounding effects of changing CarD and σ⁹³ levels in vivo. The results are shown in Supplementary Fig. 7f. Whilst CarD was able to stimulate transcription from all DNA templates, the impact of CarD was greater when supercoiling levels were higher.

## Discussion

We have discovered that many *R. sphaeroides* promoters are co-regulated by CarD and negative DNA supercoiling (Figs. 3–5). This synergy is logical; underwound DNA and CarD both facilitate duplex opening and so stability of the CarD:RNAP:DNA complex[9,12,16,18,19]. At promoters with low basal activity, and likely because DNA opening is inefficient, CarD and negative supercoiling can co-activate transcription (Figs. 3–5 and Supplementary Fig. S7). Conversely, at promoters with substantial basal activity, CarD either has little impact, or transcription is repressed (Figs. 3, 5 and 6). The −7 T mutation, which independently supports DNA opening, prevents co-operative activation by CarD and DNA supercoiling (Fig. 5). Hence, this sequence

feature is absent from most CarD-bound promoters (Fig. 1f). Surprisingly, there is no key sequence feature that distinguishes CarD activated from repressed promoters (Fig. 3e). For example, replacing key sections of the CarD dependent *rrnB*P1 promoter, with elements of the CarD-repressed P*carD*, renders *rrnB*P1 more active, and CarD independent, rather than CarD repressed. We suggest that subtle differences in promoter sequence, not necessarily conserved between different CarD targets, could be important. Indeed, in this work, changes in DNA topology, rather than sequence, allowed CarD to activate rather than repress transcription (Fig. 5).

Our conclusions above are consistent with the combined impact of CarD and DNA topology differing according to biological context and promoter kinetics. For example, levels of topoisomerase activity can differ according to the cellular environment[46]. This means that even closely related microbes can exhibit different levels of chromosomal supercoiling[47]. This may partly explain why, in *M. tuberculosis*, and *Thermus thermophilus*, CarD activates transcription from relaxed DNA templates[10–14,48]. Indeed, CarD represses the *M. tuberculosis rrnA*P3 promoter if DNA is negatively supercoiled[11,48]. This likely results from over-stabilisation of DNA opening, and reflects key differences between *R. sphaeroides* and *M. tuberculosis*[11]. For instance, the latter organism does not routinely use -10 elements lacking the key −7 T base. Furthermore, during strand separation, *M. tuberculosis* CarD

makes more extensive contacts with the DNA backbone[12,13]. Together, these distinctions indicate more barriers to DNA opening, and so an acute need for negative supercoiling in *R. sphaeroides*. We show that the bacterium exploits this to control genes required for key housekeeping functions, including rRNA synthesis (Figs. 3–7 and Supplementary Fig. 7). This is a rational strategy, as greater negative supercoiling is an indicator of rapid cell division[21,22,49]. Indeed, DNA topology similarly contributes to rRNA regulation in *Bacillus subtilis*[50].

In *E. coli*, RNAP readily forms open complexes, and the organism does not encode *carD*[12,38]. Even so, as in *R. sphaeroides*, chromosome folding impacts the expression of housekeeping genes. Briefly, fast growth coincides with high intracellular levels of Fis, the Factor for Inversion Stimulation[51]. Primarily a DNA folding protein, Fis also binds promoters and stabilises their interaction with RNAP[52–54]. In this way, Fis specifically stimulates the expression of rRNAs and other important genes[55]. Intriguingly, Fis is also known to bind sites where DNA strands cross each other, a feature of plectonemes in negatively supercoiled DNA[56]. Additionally, starved *E. coli* also undergo chromosomal relaxation[57]. Thus, whilst the details differ, both *E. coli* and *R. sphaeroides* use chromosome biology to couple gene expression to growth rate. Similarly, both organisms utilise the alarmone ppGpp to repress or activate specific gene sets as growth slows; the so called stringent response[58–60]. In *E. coli*, this requires direct interaction of ppGpp and two separate interaction sites on RNAP. The first of these is at the interface of the ω and β' subunits[60]. The second involves β' and an RNAP-binding transcription factor called DksA[59]. By comparison, the role of ppGpp in *R. sphaeroides* is poorly defined. However, the presence of ω and DksA homologues suggest similarities to the *E. coli* system[61]. To conclude, it seems likely that CarD couples gene expression to DNA supercoiling in many bacteria. The precise impact is likely to depend on the ease of promoter opening in a given biological context.

## Methods

### Strains and plasmids
Strains and plasmids used for this study are provided in Supplementary Data 6. Unless stated otherwise, *R. sphaeroides* strain 2.4.1 was grown in M22 + media at 30 °C[62]. For starter cultures, a single colony was used to inoculate 10 ml of M22 + media in a 30 ml universal tube. After incubation for 2 – 3 days, at 30 °C, starter cultures were used to inoculate 100 ml of M22 + media, in a 2-litre baffled conical flask, at an initial $OD_{600}$ of 0.05. These cultures were incubated at 30 °C and aerated with rapid shaking at 250 rpm. Anaerobic photosynthetic growth was achieved by completely filling a 50 ml vessel with freshly inoculated culture. These were incubated with stirring at room temperature, next to a lamp fitted with a 20 W 2700 K bulb. When required, novobiocin was added at a final concentration of 2 μg ml$^{-1}$, followed by 20 min of further incubation.

All molecular cloning and DNA manipulations were done with standard procedures. HiFi Gibson assembly (New England Biolabs) was used for cloning promoter fragments in pRLG770. Site directed mutagenesis was performed with the Q5 SDM kit (New England Biolabs). Oligonucleotides used are included in Supplementary Data 6.

### Mapping of global transcription using cappable-seq and RNA-seq
Total RNA was isolated from exponentially growing *R. sphaeroides* as previously done for *Bacillus subtilis*[29]. Cells were lysed using an MP Biomedical FastRNA Pro Blue Kit. Genomic DNA was removed, and RNA isolated, using a Norgen Total RNA purification plus kit. Cappable-seq libraries were prepared by Vertis Biotechnologie AG[26,27] and sequenced using an Illumina NextSeq 500 system (75 bp read length). Standard RNA-seq library preparation was done by Azenta Life Sciences, and 150 bp paired-end sequencing done using an Illumina HiSeq. Biological duplicates were analysed in all cases. Raw sequencing

data are accessible from ArrayExpress using accession code E-MTAB-15087. Cappable-seq data were processed as described previously to identify TSSs[27,29]. Briefly, FASTQ files were aligned to the *R. sphaeroides* genome (GenBank accession numbers CP000143.2 and CP000144.2) with Bowtie 2 (Galaxy Version 2.4.2) using default settings[63]. Coverage at each genomic position was calculated with the genomecov function of Bedtools (version 2.26)[64]. Chromosomal positions with a minimum read depth of 4, and at least three times higher coverage than the previous position, in both biological replicates, were called as TSSs. Read depth at each genome position was normalised to the total reads aligning to chromosome 1 and 2. The normalised read depth for each TSS for both replicates was averaged and used as a measure of activity. Using this value, all TSSs were categorised into deciles reflecting their activity. TSSs were assigned to the nearest gene less than 300 bp downstream on the same strand.

For differential gene expression analysis of RNA-seq data, read counts for each replicate were calculated from Binary Alignment Map (BAM) files using featureCounts (version 2.6)[65] and parsed with the exact function of EdgeR (Galaxy version 3.34.0)[66]. Bidirectional TSS were identified as in prior work[27,67]. Data were visualised with Integrative Genome Visualiser (IGV)[68]. Circular plots were generated using shinyCircos-V2.0[69].

### Promoter sequence analysis
To generate promoter consensus sequences, optimal, and appropriately positioned, DNA elements were identified upstream of each TSS. The 5 bp region upstream of the TSS was taken as the discriminator. For the −10 element, the closest match to the sequence 5'-TATAAT-3', separated from the TSS by between 4 and 8 bp, was used. The 17 bp upstream of the selected −10 element was taken as the spacer sequence. A −35 sequence, best matching the consensus 5'-TTGACA-3', was then selected between 16 and 18 bp upstream of the designated −10 motif. Individual consensus motifs were generated, by aligning discrete promoter regions, to account for variation in element spacing. The exception was the generation of bidirectional promoter motifs. Sequence logos were generated with Web logo 3[70]. The position weight matrix used to score different CarD-bound promoter classes is in Supplementary Fig. 4b.

### Mapping of global protein-DNA interactions using ChIP-seq
Exponentially growing *R. sphaeroides*, with or without 2 μg ml$^{-1}$ novobiocin, were crosslinked with 1% (*v/v*) formaldehyde for 20 minutes. The reaction was quenched with 0.5 M glycine, and cells were resuspended in lysis buffer (50 mM HEPES pH 7.5, 150 mM NaCl, 1 mM EDTA, 1% (*v/v*) Triton X-100, 0.1% (*v/v*) sodium deoxycholate and 1 protease inhibitor cocktail tablet per 50 ml) with 2 mg ml$^{-1}$ lysozyme. After sonication, 20 μl of chromatin was kept as an input control. Approximately 150 μg of chromatin was immunoprecipitated by the addition of anti-CarD antibodies (a gift from Wilma Ross, University of Wisconsin-Madison) or anti-σ antibodies (Biolegend, clone 2G10). Protein-DNA complexes were immunoprecipitated with a 1:1 mix of Protein A/G Dynabeads (Thermo Fisher). Before eluting complexes, beads were washed with lysis buffer, high salt (500 mM NaCl) lysis buffer, wash buffer (20 mM Tris pH 8, 250 mM LiCl, 0.5% (*v/v*) NP-40, 0.5% (*v/v*) sodium deoxycholate and 1 mM EDTA) and TE. Elution, protein degradation (1 hour incubation with proteinase K at 42 °C), and reverse crosslinking (4 hours at 65 °C) was done in elution buffer (50 mM Tris pH 7.5, 10 mM EDTA and 1% (*v/v*) SDS). DNA was purified from the eluate using a Qiaquick PCR clean up kit (Qiagen). Sequencing libraries were then prepared using the NEB next Ultra II kit (New England Biolabs). Library concentrations were determined by qPCR using a NEB next library quantification kit (New England Biolabs) and pooled before being sequenced using an Illumina NovaSeq instrument. Data is available using ArrayExpress accession code E-MTAB-15087. Reads were aligned, and coverage determined, as described above. Data were

normalised so that the average read depth per base was the same for each experiment. Peaks with a minimum width of 200 bp, and normalised read depth of 50, were called as binding sites and this also defined the peak boundaries. To determine changes in protein binding, read counts for each binding site were extracted from BAM files, using the featureCounts function of Rsubread (version 2.6.4)[71]. Read count changes were identified using EdgeR (version 3.33.7)[66].

Sites where CarD and $\sigma^{93}$ binding sites overlapped, with each other or TSSs, were identified using the subsetByOverlaps function of the Genomic Ranges package (version 1.44)[72]. Peaks were defined as overlapping if any part of one peak fell within the boundaries of the other. Peaks for CarD or $\sigma^{93}$ were associated to genes using the findOverlap function of the Genomic Ranges package with additional conditions[72]. Briefly, a CarD peak was assigned to a gene if the upstream peak boundary was upstream of the gene start codon and if the downstream boundary overlapped the coding sequence. For $\sigma^{93}$ peaks, the same rules were used. Additionally, in cases where the entirety of a short gene (e.g., for a tRNA) fell within the peak boundaries, the peak was assigned to that gene. All $\sigma^{93}$ peaks beginning inside a gene coding region were designated intragenic, if not already assigned to an overlapping gene.

## Mapping of global DNA supercoiling using Psora-seq
Experiments were done as described by Visser et al. with the following changes[35]. Genomic DNA from exponentially growing cells was extracted with CTAB resuspended in 225 µl of water. Genomic DNA was sheared to a size of 400 – 500 bp using a Biorupter (Diaganode). 50 µl of sheared genomic DNA was kept as an input sample, and the remainder was immobilised on streptavidin T1 dynabeads (Thermo-Fisher). After washing and reverse crosslinking as per Visser et al, eluted DNA was precipitated and resuspended in 20 µl water. Library preparation and sequencing was done by Azenta Life Sciences. Sequencing was done using an Illumina NovaSeq in 150 bp paired-end configuration. Data is available using the ArrayExpress accession code E-MTAB-15087. FASTQ files were trimmed using trimmomatic to remove Illumina adaptors (TruSeq 3 setting) and hard trimmed to keep the first 65 bp. Trimmed reads were aligned to the genome as described above. Read counts for 500 bp genomic windows were determined using the multiBamSummary package of deepTools2 (Galaxy Version 3.5.4)[73]. Windowed read counts were normalised using a scaling factor (of 1) divided by the average window read count. Each window's psoralen binding value was calculated by taking the $\text{Log}_2$ of the normalised read count divided by the normalised input read count. Psoralen binding values derived from Psora-seq windows that overlapped CarD binding sites were determined using the subsetByOverlaps function of the Genomic Ranges package (version 1.44)[72]. Circular plots were generated using shinyCircos-V2.0[69]

## Relaxation of plasmid DNA using topoisomerase I
Negatively supercoiled plasmid DNA (5 µg) was treated with 10 units of calf thymus topoisomerase I (Takara Biosciences) in 50 µl of the provided reaction buffer for 90 minutes at 37 °C. If required, ethidium bromide was added at concentrations indicated. Reactions were stopped by the addition of phenol:chloroform:isoamyl alcohol (25:24:1), and DNA was recovered from the aqueous phase by ethanol precipitation. Plasmid isoforms (200 ng) were separated using 1% (v/v) agarose gels, containing ± 5 µg ml$^{-1}$ chloroquine. DNA bands were visualised by post-staining with ethidium bromide (~1 µg ml$^{-1}$).

## ChIP-qPCR
Chromatin was immunoprecipitated as above from 40 ml R. sphaeroides grown aerobically to the exponential or stationary phase in M22 + media. Quantitative PCR was performed using Luna Universal qPCR Master Mix (New England Biolabs) on an Agilent AriaMX system, with primer pairs shown in Supplementary Data 6. Each sample had 3

technical replicates and was one of 2 biological replicates. Enrichment was calculated as % input, with Ct values of input control being scaled to 100%.

## RT-qPCR
Total RNA was purified from exponential and stationary phase R. sphaeroides cultures as above. Cell mass approximately equivalent to 10 ml at OD$_{600}$ 0.5 was used. Total RNA was treated with DNase I (Zymogen) before clean-up with a Qiagen RNeasy kit. RT-qPCR was performed using the Luna Universal Probe One-Step RT-qPCR kit (New England Biolabs) on an Agilent AriaMX system. Primers are included in Supplementary Data 6. Fold enrichment was calculated using the $2^{-\Delta\text{CT}}$ method, normalised to the constitutively expressed PutR control.

## Purification of R. sphaeroides RNA polymerase
R. sphaeroides $\sigma^{93}$ RNAP was purified from strain 2.4.1 encoding a C-terminally His$_{10}$ tagged β' subunit (a gift from Wilma Ross and Rick Gourse, University of Wisconsin–Madison)[9]. Cultures were grown in LB medium and cells harvested in exponential phase (OD$_{600}$ ~ 0.5). Pellets were resuspended in 20 ml of BugBuster (Millipore) protein extraction reagent diluted to the working concentration in Ni$^{2+}$ buffer (20 mM Tris pH 7.9, 600 mM NaCl and 5 % (v/v) glycerol). Resuspended cells were supplemented with 1 cOmplete EDTA-free protease inhibitor tablet (Roche), 4 mg of lysozyme, ~150 units of DNase I (Roche), and 35 units of RNase A (Qiagen) before room temperature incubation with gentle agitation for 1 hour. The resuspension was sonicated for 2 min (30 s on/off, 25 % amplitude) before clarification by centrifugation (45,000 x g for 20 min at 4 °C) and filtration (0.2 µM PES filter). Lysate was loaded onto a His-trap HP column (Cytiva) pre-equilibrated with Ni$^{2+}$ buffer. Unbound and weakly bound proteins were removed by washing with 8 column volumes of Ni$^{2+}$ buffer and then Ni$^{2+}$ buffer + 25 mM imidazole. Bound proteins were eluted with stepwise washes with 50, 100, 200 and 300 mM imidazole in Ni$^{2+}$ buffer. RNAP was eluted in 200 mM and 300 mM imidazole fractions, which were pooled and concentrated with a 50 KDa Vivaspin concentrator (Merck). Concentrated protein was dialysed overnight into 2 x storage buffer (40 mM Tris pH 7.9, 400 mM KCl, 2 mM DTT, 0.2 mM EDTA) before further concentration. An equal volume of 100 % glycerol was added before storing the protein at −20 °C.

## Purification of R. sphaeroides CarD protein
The pET SUMO plasmid for expression of an N terminally His$_{10}$ – SUMO tagged CarD was gifted by Dr Wilma Ross and Prof Rick Gourse (University of Wisconsin–Madison)[9]. E. coli T7 Express cells (New England Biolabs) carrying the plasmid were grown in LB at 37 °C until the exponential phase, at which point cells were shifted to 18 °C, induced with 1 mM IPTG and incubated overnight. Cell lysis and Ni$^{2+}$ chromatography was performed as above. 200 mM and 300 mM imidazole fractions containing CarD were pooled and concentrated. To remove His$_{10}$-SUMO tag, 40 µg Ulp1 protease was added to ~ 12 mg CarD and dialysed overnight at 4 °C into Ulp1 buffer (20 mM Tris pH 7.9, 150 mM NaCl and 1 mM β-mercaptoethanol). Precipitate was removed through filtration (0.45 µM PES filter) and protein was passed over His-trap HP column as before, except tag-free CarD was present in flow through. CarD was concentrated with a Vivaspin concentrator before the addition of an equal volume of 100 % glycerol for storage at −20 °C.

## Western blotting
R. sphaeroides cells, equivalent to a suspension having an OD$_{600}$ of 0.5, were pelleted and resuspended in 200 µl lysis buffer (1 x Bug Buster (Merck), 20 mM Tris HCl pH 7.9, 600 mM NaCl and 5% (v/v) glycerol) and incubated with 1 µl of Lysonase (Merck) for 20 minutes at room temperature before 10 cycles of sonication on a Bioruptor (Diagenode). For each sample, 5 µg of lysate was resolved by SDS-PAGE before transfer to nitrocellulose. After blocking, anti-CarD (a gift from

Wilma Ross, University of Wisconsin-Madison) or anti-σ factor primary antibodies were used at 1:5000 and 1:1000 dilutions, respectively. Secondary antibodies, anti-rabbit HRP (Thermo-fisher Scientific. cat no 31464) for CarD and anti-mouse HRP (Sigma-Aldrich. cat no 12-349) for σ factor, were used at 1:10 000 dilutions. Western blots were visualised with the SuperSignal™ West Pico PLUS ECL substrate kit (Thermofisher).

### In vitro transcription assays

In vitro transcription reactions contained 0.01 μM DNA template, 0.05 μM $\sigma^{93}$ RNAP and, where applicable, 2 μM of CarD in transcription buffer (20 mM Tris pH 7.9, 170 mM NaCl, 10 mM $MgCl_2$, 1 mM DTT and 0.1 mg ml$^{-1}$ BSA). Reactions were initiated by the addition of an NTP mix (200 μM ATP/GTP/CTP, 10 μM UTP and 2 μCi [α-$^{32}$P] UTP) and incubated for 20 minutes at 30 °C. Reactions were stopped by the addition of 30 μl acidic phenol (pH 4.3). After centrifugation, the aqueous phase was added to an equal volume of formamide containing stop solution, before separating transcripts using an 8% (w/v) denaturing polyacrylamide gel. Gels were exposed overnight using Bio-Rad phosphor screens, and signals visualised with a Bio-Rad Personal Molecular Imager, using Quantity One software (version 4.6.9). Plasmid template DNA was purified from NEB 5-α E. coli cells (New England Biolabs) using ZymoPURE II Plasmid Midiprep Kit (Zymogen). For quantification, band densities were measured using Bio-Rad Image Lab (version 6.1). The value for reactions with CarD was divided by the same value for the corresponding reaction without CarD to give the fold activation by CarD.

### Analysis of gene ontology

Predicted COG terms for all R. sphaeroides encoded proteins were obtained from EggNOG-mapper, which assigns terms based on orthology[74]. Assigned COG terms were manually curated, with unannotated proteins designated as S for function unknown. Non-coding RNA (tRNA, ncRNA, SRP RNA and rRNA) were added manually as separate terms.

### Reporting summary

Further information on research design is available in the Nature Portfolio Reporting Summary linked to this article.

## Data availability

Illumina sequencing data are available from Array Express using the accession code E-MTAB-15087. Results generated from the analysis of sequencing data, such as differential gene expression analysis, are available in Supplementary Data 1–5. Source data are provided in this paper.

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

## Acknowledgements

We thank Tony Maxwell for the protocols and Steve Busby for helpful conversations. We are grateful to Rick Gourse, Wilma Ross, Kemardo Henry, Andrew Hitchcock and Joe Wade for providing materials, advice, and commenting on the manuscript prior to submission. This work was funded by Wellcome Trust Investigator Award (no. 212193/Z/18/Z) and Leverhulme Trust programme grant (no. RPG-2024-168) to DCG.

## Author contributions

D.F. and D.C.G. designed the project. All experimental work was done by D.F. with support and advice from E.A.W. Bioinformatics was done by D.F. and D.C.G. The paper was written by D.F. and D.C.G. with all authors making edits as appropriate.

## Competing interests

The authors declare no competing interests.
