## [Transparent Peer Review file · Nature Communications]

The bacterial RNA polymerase-associated CarD protein couples promoter activity to DNA supercoiling

Corresponding Author: Professor David Grainger

Version 0:

Reviewer comments:

Reviewer #1

(Remarks to the Author)

In this manuscript, Forest et al. employed a combination of genome-wide techniques (RNA-seq, ChIP-seq, Cappable-seq, and psoralen-seq), along with in vitro transcription assays, to investigate the interplay between CarD and DNA supercoiling, and how they together regulate promoter activity in *Rhodobacter sphaeroides*. The authors propose that CarD “couples housekeeping transcription to the environment via DNA topology.”

I find the evidence convincing that CarD indeed links promoter activity to DNA supercoiling. However, the claim that CarD couples transcription to the environment via changes in DNA topology is more speculative. This would be more compelling if demonstrated under physiologically relevant conditions, such as stationary phase, rather than through novobiocin treatment alone. During stationary phase (or slower growth conditions/media), negative supercoiling is expected to decrease, which could provide a suitable context for testing whether CarD mediates environmentally responsive transcription. One caveat is that CarD level is reduced during stationary phase in *R. sphaeroides* (I assume the *carD* promoter itself might be supercoiling-sensitive?)

A question remains incompletely resolved: how are CarD, the frequent absence of a -7T position, and negative supercoiling mechanistically linked? Is CarD-DNA binding intrinsically sensitive to DNA topology? The ChIP-seq data suggest so, but can this be tested in vitro? Given the “tryptophan wedge” mechanism proposed for CarD in *Mycobacterium tuberculosis*, can the authors speculate on how *Rhodobacter* CarD, RNA polymerase, and supercoiling might function together mechanistically.

Expanding the Discussion, together with a “model” figure (Figure 7?), would greatly help readers.

What were the controls used in the ChIP-seq experiments? Polyclonal antibodies against CarD and σ^{93} were used for immunoprecipitation, but polyclonal antibodies are notoriously prone to nonspecific binding. Since *carD* and *rpoD* are essential genes, deletion controls are not feasible. However, *R. sphaeroides* is genetically tractable, so epitope-tagging could be an alternative, though I acknowledge it might require substantial effort. Including some control experiments to address antibody specificity would help reassure readers.

Additionally, can a consensus DNA-binding motif for CarD be derived from the ChIP-seq data?

Regarding psoralen-seq, each bin spans 5 kb and could contain multiple genes or promoters. How reliable is it to assign a CarD-dependent promoter to such broad bins?

The presentation of data throughout the manuscript could be improved. As it stands, the manuscript is mainly for aficionados in bacterial transcription field i.e. not accessible for a wider readership. For example, in Figure 1f, the x-axis should be labeled using the conventional +1, -10, and -35 nomenclature, rather than sequential positions from 1 to 60. It would be helpful to annotate, for example the -7 position as well. I recommend adopting the labeling style used in Henry et al. (PNAS, 2020; reference 6).

In Figure 3d (left panel), some CarD-associated promoters are upregulated following novobiocin treatment, rather than downregulated as one might expect. These cases are intriguing: what are these promoters, and what are their distinguishing features? A deeper analysis could yield additional insights into the relationship between CarD and supercoiling, yet this was not discussed/investigated in the manuscript.

Finally, in Figure 3c, the manuscript refers to changes in ChIP-seq peak height as “upregulated” or “downregulated.” I assume these terms are reserved for transcriptomic data?

Also, eliminate spaces between numbers and percent signs (e.g., use “50%” instead of “50 %”).

Reviewer #2

(Remarks to the Author)

This study offers compelling observations on the roles of CarD and DNA supercoiling in transcriptional regulation in *R. sphaeroides*. However, several aspects of data interpretation require closer alignment with existing mechanistic and structural work. In particular, a more thorough analysis of promoter architecture—including the -35 and extended -10 motifs—alongside consideration of non-canonical initiation mechanisms such as post-termination complexes (PTCs), is essential. The integration of ChIP-seq, Psoralen-seq, RNA-seq, and cappable-seq represents a significant strength, enabling a genome-wide view of regulatory dynamics. Together, these approaches provide complementary insights into promoter activity, DNA topology, and protein-DNA interactions. With the revisions and clarifications outlined above, this work could make a substantial contribution to our understanding of transcriptional regulation in bacteria. This work presents exciting observations but the interpretations need to be grounded more firmly in the mechanistic literature. A more rigorous analysis of promoter architecture (especially the -35 and extended -10 motifs), consideration of non-canonical initiation via PTCs, and improved data presentation will significantly strengthen the manuscript.

Major Points for Revision:

1. -35 element and promoter assignment (fig. 1):

The lack of -35 element analysis is a puzzling. Why is this not included in the assignment of bona fide promoters? *R. sphaeroides* appears to have canonical -35 elements and lacks RbpA (PMID: 33168725), so their inclusion in promoter analysis is essential.

2. clarification of classification in cappable-seq:

It is unclear whether the cappable-seq approach filters out transcript reads without recognizable promoter elements. How are genic and intergenic reads defined by the authors?

3. Interpretation of intergenic/nongenic transcription in context of post-termination complexes (PTCs):

Much of the cappable-seq signal appears intergenic. There is a growing body of literature showing that RNAP can initiate transcription without sigma via post-termination complexes (PTCs) (PMIDs: 39779919, 37406096, 37459525). These initiate at -10-like elements without -35 motifs and may recruit sigma and CarD (PMID: 31974358). The authors must consider:

-Whether some of the observed TSSs originate from PTCs rather than bona fide promoters.

-The presence of bidirectional symmetry and absence of -35 motifs as features of PTC-derived initiation.

-That PTCs are highly sensitive to supercoiling (PMID: 39779919).

-That *R. sphaeroides* appears to encode a RapA-like factor (WP_119001171.1), which clears PTCs.

That CarD and sigma association with intergenic TSSs may reflect stabilization of PTC bubbles rather than standard promoter interactions.

4. Missing -35 and extended -10 motif analysis:

Explicit analysis of these motifs would help determine which TSSs are likely sigma-dependent versus PTC-derived. Without it, many TSSs may be misclassified.

5. Mechanistic interpretation of CarD and supercoiling (including Fig. 4C):

Recent findings in *M. tuberculosis* suggest that CarD's effects are inhibited on negatively supercoiled DNA (PMID: 40060575). The authors should:

-Discuss this preprint in more detail and reconcile the discrepancy with their findings in a more mechanistic way.

-Clarify that novobiocin reduces negative supercoiling, likely destabilizing RPo indirectly, thus complicating their conclusions.

-Consider the potential impact of novobiocin on PTC activity.

-Quantify transcription in all lanes in Fig. 4C to compare CarD+ vs. CarD- at increasing negative supercoiling levels.

-Discuss why RNAI is used as a control and provide its promoter sequence—CarD has no apparent effect here, which needs explanation.

-More clearly quantify results from Figs. 4 and 5 to support conclusions about the effects of -10 element mutations.

6. Analysis and rigor (Figs. 4 and 5):

-Indicate the number of replicates for each experiment.

-Quantify transcription in all lanes in Fig. 4C to compare CarD+ vs. CarD- at increasing negative supercoiling levels. The current quantification in 4c is just CarD. Also where is the error bar in the red bar?

-Provide P-values and statistical significance.

-In the supplement, label each gel lane and replicate clearly (e.g., replicate 1, 2, 3).

7. Mechanisms (Fig. 3D and Line 143):

-The statement that CarD depletion causes "open complex collapse" should also consider that it might promote faster

escape into elongation.

-Novobiocin treatment results in both upregulation and downregulation of transcription—even at CarD-associated promoters—this duality needs discussion. It is currently ignored.

- Different promoters may respond to CarD in different ways depending on their kinetic properties (PMID: 37075846); this should be acknowledged.

8. Discussion:

The discussion should incorporate:

-The broader literature on CarD function, supercoiling effects, initiation kinetics, and promoter escape.

-The likely contribution of topoisomerases and DNA topology to transcription across bacteria.

-A critical evaluation of whether all observed TSSs represent bona fide promoters.

-The roles of DksA seem less relevant and can be de-emphasized in favor of the points above.

Reviewer #3

(Remarks to the Author)

Forrest et al is a well-designed study on the effects of the transcription factor CarD in regulating *Rhodobacter sphaeroides* promoters. The authors combine various high-throughput data, including ChIP-seq, RNA-seq, transcript start site mapping (cappable-seq), and negative supercoiling mapping (with psoralen sequencing) to define the CarD regulon and supercoiling-sensitive genes of *R. sphaeroides*. These authors then use biochemical experiments with plasmids of different negative supercoiling to demonstrate that the *rrnBP1* promoter requires both CarD and negative supercoiling for activation, with this requirement bypassed by a -7A → T transversion in the -10 sequence. The authors thus conclude that “an optimal -10 element sequence, negative supercoiling, and CarD all enhance promoter activity, and that maximal *R. sphaeroides* transcription requires only two of these stimulatory factors.”

I found the global study technically sound and the biochemical experiment especially compelling. Although a relationship between CarD, supercoiling, and open complex formation has been reported, the high-throughput studies are novel in *R. sphaeroides* and the biochemical experiment clearly demonstrates the relationship between CarD and negative supercoiling at the *rrnBP1* promoter. However, these studies are not sufficiently complementary, with the conclusions are not justified by the experiments. I believe that the authors are conflating multiple distinct classes of CarD-dependent promoters in this study. I also have many questions about the proposed model and experimental approaches. Either more experiments or significantly more limited interpretations of the data is warranted.

Major concerns:

1) As noted by the authors, CarD-dependent promoters can be repressive or stimulatory, but no effort is made to distinguish between these classes. Indeed, previous work in *Mycobacteria* suggests that CarD-mediated open complex formation could either promote promoter escape (activation) or inhibit it (repression) (Zhu et al, PMID: 31217290), suggesting that CarD/supercoiling could identically impact open complex formation at many promoters but cause vastly different transcriptional outcomes.

a. Only the *rrnBP1* promoter is tested biochemically and it is not made clear the relationship between this promoter and the high-throughput experiments. To what extent is *rrnBP1* an appropriate model for the relaxation-repressed, CarD-bound promoter class proposed in this study? Is the *rrnBP1* promoter bound by CarD in ChIP-seq and significantly downregulated in response to chromosome relaxation? What is the sequence of the *rrnBP1* promoter and how does it relate to the predicted CarD-dependent promoters (see further comments regarding sequence logo in 1b, below)? Biochemically validating additional CarD-associated novobiocin-repressed promoters is essential to strengthen the conclusions presented here.

It is also clear from this study and the literature that there is a class of CarD-repressed promoters, with the prime example the CarD promoter itself (Henry et al, PMID: 34152199). Henry et al showed that an extended -10 element and other spacer sequences mediated repression. In this work, the authors have a set of CarD-dependent relaxation-activated promoters (Figure 3) that are overlooked. Do these promoters have the same conserved sequences as those that are relaxation-repressed? Is the CarD promoter supercoiling-dependent and is it repressed or activated? Are the sequence features identified in Henry et al observed in any promoter classes? An ideal experiment would be to deplete CarD and reexamine transcription to distinguish between these promoter sets but some effort should be made to categorize and validate that the promoter features associated with different CarD/supercoiling regulation classes. The authors should also consider biochemically testing CarD-associated novobiocin-induced promoters. In these cases, is CarD functioning in synergy or opposition to supercoiling? Without these experiments, the authors should be extremely cautious in generalizing their biochemical result to encompass the full range of possible CarD/supercoiling interactions.

b. The authors make an argument for -7T/-12T de-enrichment in CarD promoters. However, there are several issues with idea. In figure 1F, I cannot see the difference in the -7T/-12T as reported. The de-enrichment of -7T is well-supported by the literature (Henry et al, PMID: 33168725; 34152199) but T is still an enriched base in the sequence logo presented. The -12 T is clearly enriched as well. Is the interpretation based on pvalue? It does not make sense to quantitatively compare two p-values. I would suggest the bigger differences are that CarD-bound promoters lack -8T and a G at -33. Perhaps a quantitative analysis like in Henry et al (PMID: 33168725) would be more appropriate.

More worryingly, nothing is ever explored for -12T (or for any other promoter differences). Henry et al (PMID: 33168725) observed that -12T (without -7T) was conserved for 15 promoters activated by CarD in vitro. Further, the CarD promoter itself contains a -12T. What is the explanation for these discrepancies? What is the function, if any for these sequences? How do

they relate to the different potential CarD/supercoiling promoter classes (1a, above)? Does -12T have any effect on CarD/negative supercoiling synergy?

At the end of the work, I am convinced that for *rrnBP1* both CarD and negative supercoiling are required for activation. However, I have many remaining questions about the overall model.

2) Why is *rrnBP1* be regulated in this fashion? I agree with the authors that synergy is logical, but this operates more like an “AND” logic gate, meaning the effects are not tunable. Is it to quickly shutoff ribosomal synthesis during unfavorable conditions? I feel this idea would be better-supported if there are additional examples of this regulation. Perhaps this lack of synergy is unique to *rrnBP1*.

3) How generalizable is this finding for all *R. sphaeroides* promoters? I also agree that “the combined impact of CarD and DNA topology must differ according to biological context”. Examining several more promoters and, ideally, in different regulatory classes (novo-activated, novo-inhibited, CarD-bound, CarD-free, etc) is essential for extending the experiments to infer CarD’s effects on individual promoters. This is especially true since CarD is itself is controlled by a negative feedback loop. Whether negative supercoiling at the CarD promoter increases or represses CarD expression will have consequences for the interpretation of the transcriptomic/ChIP data, especially if CarD is rapidly turned over.

4) How does CarD/RNAP together sense negative supercoiling? Figures 2-3 show that CarD-bound sites are more negatively supercoiled than random sites. However, are these CarD-bound sites more negatively supercoiled than $\sigma 93$ -bound sites? One would imagine that any active TSS must be negatively supercoiled since negative supercoiling promotes promoter melting and RNAP escape. If CarD-free $\sigma 93$ -bound sites are more negatively supercoiled, CarD could perhaps compensate for the absence of negative supercoiling at promoters rather than require it for binding, which is the model presented here. This analysis should also be performed again in the novobiocin-treated dataset, as it would be interesting to know if $\sigma 93$ /RNAP remains more bound when CarD is present/absent, which would support the hypothesis that CarD stabilizes open complexes.

Relatedly, for Figure 6B, because CarD is associated with $\sigma 93$ (housekeeping) promoters, this set should be naturally enriched in metabolism and central dogma processes. I’d be curious to see the comparison to the remaining $\sigma 93$ -dependent non-CarD promoters, which are also likely enriched for these functions. Significant differences between these two sets would be more compelling than the comparison to random promoters, which are likely enriched for alternative sigma factor promoters that regulate stress response genes.

5) Based on the Methods, it is not clear how the $\sigma 93$ and CarD overlapping promoters were selected. Further definition of what “overlap” constitutes would be helpful. I also wonder where CarD binds at these promoters (e.g. around TSS, -10, or hundreds of bases upstream) and if this information could be used to refine the models for CarD/supercoiling behaviors.

6) Is there a relationship between CarD and the bidirectional promoters noted in Figure 1? Is the hypothesis that CarD should activate these promoters and it should activate both? Is CarD bind to these bidirectional promoters?

7) Line 147: Is the idea here that more CarD-bound genes are downregulated than CarD-independent? Can this be explained more clearly (e.g. more CarD-bound genes were downregulated)? There also appears to be more genes upregulated for CarD-independent promoters, is this correct?

8) Line 151-154: I did not understand the data for “Novobiocin predominantly downregulates transcription and this is more pronounced for CarD bound genes”. Is the idea that more CarD-bound genes are downregulated upon treatment? To my eye, it appears that there are more genes downregulated for CarD-free promoters. I also suggest the statement “novobiocin caused significant DNA relaxation, loss of both CarD and $\sigma 93$ binding, and lower gene expression” should be revised. Novobiocin-induced/DNA relaxation-induced genes have been observed in diverse organisms, and there are many genes (red) that are induced upon DNA relaxation in Figure 3D.

Minor comments:

- 1) The authors should carefully check over their referencing and reference the primary literature as often as possible:
 - a. Line 27-28: a review here would be more appropriate, as reference 2 (Zhang et al) were only able to visualize domains 2-4 and do not discuss the role of sigma domains in depth. Reference 3 also does not the primary source to show sigma and -10/-35 contacts.
 - b. Line 30: Instead of references 2-4, which are not the references that show sigma domain 2 interacting with -7T/-11A, please cite Feklistov and Darst (PMID: 22136875), Zhang et al (PMID: 23086998), and Zuo and Steitz (PMID: 25866247).
 - c. Line 108: The authors may also want to cite Lal et al (PMID: 27025941), the first report of psoralen use genome-wide in bacteria.
- 2) Line 46-47: The authors are correct that lack of rotation of RNAP increases supercoiling. However, simple duplex DNA melting by RNAP also leads to torsional strain and consequent twisting of DNA.
- 3) Line 75: Is there some way to distinguish between “active” and “infrequently used” promoters from cappable-seq? Perhaps the studies would provide better motif resolution if the highest-active promoters are selected.
- 4) Please renumber the logo (indicating +1) in Figure 1F and mark the correct bases to examine to ease the burden on the reader.

Version 1:

Reviewer comments:

Reviewer #1

(Remarks to the Author)

The authors have address my concerns, no further comment. Thank you.

Reviewer #2

(Remarks to the Author)

The authors have satisfactorily addressed my previous comments. My only remaining concern is the use of the word “flaw” throughout the manuscript (including the Abstract and Introduction) to describe the lack of consensus among promoter elements. I recommend replacing this term with more precise language—for example, explicitly stating that certain promoters deviate from the canonical –10 consensus or lack specific consensus features such as the –7T. The term “flaw” implies a subjective judgment, whereas a more descriptive formulation would be more precise.

Reviewer #3

(Remarks to the Author)

I commend the authors on the extensive and thorough revision. All the new experiments and analyses greatly improve the manuscript, and I have no further comments. Congratulations on the fine work.

We thank the reviewers for their considerate assessment of our manuscript. In the vast majority of cases, we agree with the reviewers and have taken their suggestions on board when revising our paper. Thus, the updated paper includes substantial new experimental work and a deeper analysis of the existing genomic data.

REVIEWER COMMENTS

Reviewer #1 (Remarks to the Author):

In this manuscript, Forest et al. employed a combination of genome-wide techniques (RNA-seq, ChIP-seq, Cappable-seq, and psoralen-seq), along with *in vitro* transcription assays, to investigate the interplay between CarD and DNA supercoiling, and how they together regulate promoter activity in *Rhodobacter sphaeroides*. The authors propose that CarD “couples housekeeping transcription to the environment via DNA topology.”

I find the evidence convincing that CarD indeed links promoter activity to DNA supercoiling. However, the claim that CarD couples transcription to the environment via changes in DNA topology is more speculative. This would be more compelling if demonstrated under physiologically relevant conditions, such as stationary phase, rather than through novobiocin treatment alone. During stationary phase (or slower growth conditions/media), negative supercoiling is expected to decrease, which could provide a suitable context for testing whether CarD mediates environmentally responsive transcription.

We have done additional stationary phase experiments that the reviewer suggests, testing binding of CarD using ChIP-qPCR and levels of transcription using RT-qPCR. Both binding of CarD, and transcription from the CarD activated promoters, is greatly reduced in stationary phase as expected. These data are presented in Figure S7a and S7b.

One caveat is that CarD level is reduced during stationary phase in *R. sphaeroides* (I assume the carD promoter itself might be supercoiling-sensitive?)

The reviewer is correct, CarD levels do decrease in stationary phase. Most likely, CarD is degraded. We also find that σ^{93} levels decrease substantially in starved cells (Figure S7c and S7d). As the reviewer notes, such changes make the above result difficult to interpret. Hence, we are cautious in our presentation of these data, noting that both supercoiling changes, and reduced levels of CarD/ σ^{93} , must contribute.

With this in mind, we have done an additional experiment that we think is more informative with respect to physiologically relevant changes in supercoiling. After comparing several different conditions, we found clear changes in plasmid supercoiling in *R. sphaeroides* cells grown aerobically compared to those dependent on photosynthesis. Rather than investigate gene expression effects *in vivo*, where many other regulatory factors are at play, we used plasmids isolated from cells grown in each condition (and so having different levels of negative supercoiling) as templates for *in vitro* transcription. The data show that CarD readily activates transcription from the more negatively supercoiled plasmid obtained from aerobically growing cells. By comparison, CarD activated transcription poorly from plasmid templates obtained from photosynthetically growing cells. Hence, physiologically relevant changes in supercoiling do impact the ability of CarD to control transcription. The data are in Figures S7e and S7f.

A question remains incompletely resolved: how are CarD, the frequent absence of a -7T position, and negative supercoiling mechanistically linked? Is CarD-DNA binding intrinsically sensitive to DNA topology? The ChIP-seq data suggest so, but can this be tested *in vitro*? Given

the "tryptophan wedge" mechanism proposed for CarD in *Mycobacterium tuberculosis*, can the authors speculate on how *Rhodobacter* CarD, RNA polymerase, and supercoiling might function together mechanistically.

Expanding the Discussion, together with a "model" figure (Figure 7?), would greatly help readers.

We apologise for the mechanistic link not being clear. Negative supercoiling (by virtue of the underwound double helix) CarD (via its tryptophan wedge) and -7T (due to base flipping and σ binding) all contribute to promoter open complex stability, and this is the mechanistic link. Thus, at the *rrnB* promoter, which lacks -7T, both CarD and negative supercoiling are needed for transcription to occur. If the -7T base is introduced by mutation, it is possible to dispense with either CarD or negative supercoiling.

We don't think CarD binding is intrinsically sensitive to DNA topology. Rather, if a stable open complex can't be formed (e.g. because of a reduction in negative supercoiling) then the entire transcriptional apparatus (including CarD) is likely to vacate the promoter DNA prematurely.

As requested, we have added extra text. This appears at the end of Results sections "Decreased DNA supercoiling impacts CarD..." and "Reduced DNA supercoiling predominantly inhibits...". We also now mention this in the first paragraph of the Discussion. At this point we haven't included a "model" figure. We tried several graphic representations but found it difficult to simultaneously show the different regulatory outcomes.

What were the controls used in the ChIP-seq experiments? Polyclonal antibodies against CarD and σ^{93} were used for immunoprecipitation, but polyclonal antibodies are notoriously prone to nonspecific binding. Since *carD* and *rpoD* are essential genes, deletion controls are not feasible. However, *R. sphaeroides* is genetically tractable, so epitope-tagging could be an alternative, though I acknowledge it might require substantial effort. Including some control experiments to address antibody specificity would help reassure readers.

We always do mock immunoprecipitations (without antibody) for ChIP-seq experiments and that was also the case here. These controls yield little DNA, consistent with specific isolation of DNA binding proteins when antibodies are present. Furthermore, the specific regions of enrichment for σ^{93} and CarD binding are not evident in the control experiment (see below). Note that the sequence reads from the control experiment are available on array express along with the ChIP-seq data for σ^{93} and CarD binding.

anti- σ^{93}

anti-CarD

no antibody

The reviewer is correct that the anti-CarD is polyclonal. To reassure the reviewer, we have done additional western blotting experiments. The anti-CarD is highly specific with no evidence of binding to other proteins. The new data are in Figure S7c.

The anti- σ^{93} is monoclonal and the epitope is precisely mapped. This is a commercially available antibody, that is highly specific and reacts with the primary sigma factors of numerous bacteria (the antibody was raised against *E. coli* σ^{70}). Almost all published ChIP-seq studies of primary sigma factors have used this antibody. Figure S7d shows western blotting experiments using this antibody. The two more mobile bands detected are likely to be degradation products (σ factors are prone to proteolysis at the linker regions between domains) since their abundance precisely matches that of full length σ^{93} throughout the growth phases.

Additionally, can a consensus DNA-binding motif for CarD be derived from the ChIP-seq data?

No, and this is expected based on structural work showing that CarD makes very minimal contacts with the DNA. Rather, CarD predominantly interacts with the RNAP.

Regarding psoralen-seq, each bin spans 5 kb and could contain multiple genes or promoters. How reliable is it to assign a CarD-dependent promoter to such broad bins?

Promoters can reliably be assigned to the correct bin, but we agree resolution is a drawback of psoralen-seq. However, in the absence of other *in vivo* approaches, our experimental options are limited. This is why we have done extensive *in vitro* studies to back up the general trends that evident from our genome-scale experiments. Given that the *in vivo* and *in vitro* data are perfectly consistent, this seems a reasonable approach. We note that, *in vivo*, supercoiling is highly dynamic and must vary between cells at any given moment. As such, the genome-scale experiments are only intended to provide a population averaged snapshot of general supercoiling trends. If supercoiling is averaged in bins that are too small, the results are likely to be less reliable.

The presentation of data throughout the manuscript could be improved. As it stands, the manuscript is mainly for aficionados in bacterial transcription field i.e. not accessible for a wider readership. For example, in Figure 1f, the x-axis should be labeled using the conventional +1, -10, and -35 nomenclature, rather than sequential positions from 1 to 60. It would be helpful to annotate, for example the -7 position as well. I recommend adopting the labeling style used in Henry et al. (PNAS, 2020; reference 6).

We have made the requested changes, now using both numbering (with respect to +1) and labelling of the key elements. The -7T position is highlighted by an asterisk and this is noted in the figure legend.

In Figure 3d (left panel), some CarD-associated promoters are upregulated following novobiocin treatment, rather than downregulated as one might expect. These cases are intriguing: what are these promoters, and what are their distinguishing features? A deeper analysis could yield additional insights into the relationship between CarD and supercoiling, yet this was not discussed/investigated in the manuscript.

We have done substantial further experiments with the *carD* promoter that is subject to repression by CarD (Figures 5, 6 and S6). We have also defined motifs for CarD-bound promoters activated and repressed by DNA relaxation (Figures 3f and 3g). There is surprisingly little difference. However, we do show that DNA supercoiling levels can determine if CarD acts

as a repressor or activator (Figure 5b). We also show that improving CarD repressed promoters gradually negates the ability of CarD to repress transcription.

Finally, in Figure 3c, the manuscript refers to changes in ChIP-seq peak height as “upregulated” or “downregulated.” I assume these terms are reserved for transcriptomic data?

We have altered the wording to instead describe increased or decreased binding.

Also, eliminate spaces between numbers and percent signs (e.g., use “50%” instead of “50 %”).

These spaces have been removed.

Reviewer #2 (Remarks to the Author):

This study offers compelling observations on the roles of CarD and DNA supercoiling in transcriptional regulation in *R. sphaeroides*. However, several aspects of data interpretation require closer alignment with existing mechanistic and structural work. In particular, a more thorough analysis of promoter architecture—including the -35 and extended -10 motifs—alongside consideration of non-canonical initiation mechanisms such as post-termination complexes (PTCs), is essential. The integration of ChIP-seq, Psoralen-seq, RNA-seq, and cappable-seq represents a significant strength, enabling a genome-wide view of regulatory dynamics. Together, these approaches provide complementary insights into promoter activity, DNA topology, and protein-DNA interactions. With the revisions and clarifications outlined above, this work could make a substantial contribution to our understanding of transcriptional regulation in bacteria. This work presents exciting observations but the interpretations need to be grounded more firmly in the mechanistic literature. A more rigorous analysis of promoter architecture (especially the -35 and extended -10 motifs), consideration of non-canonical initiation via PTCs, and improved data presentation will significantly strengthen the manuscript.

Major Points for Revision:

1. -35 element and promoter assignment (fig. 1):

The lack of -35 element analysis is a puzzling. Why is this not included in the assignment of bona fide promoters? *R. sphaeroides* appears to have canonical -35 elements and lacks RbpA (PMID: 33168725), so their inclusion in promoter analysis is essential.

We are a little confused here and apologise if we have confused the reviewer. The entire promoter (including the -35 region) was analysed and the -35 element is evident (see original Figure 3f below, with additional annotation of key features for clarity). We wonder if the reviewer is referring to the fact that the -35 element seems poorly conserved or incomplete? If so, this is always the case when promoters are analysed in this way, regardless of the bacterial species (e.g. PMID: 27806702). There are three well-known reasons for this.

a. The promoters are aligned based on TSS position. As the distance between the TSS and -35 element is not the same for every promoter (because of variation in promoter spacer and discriminator length) the final sequence logo does not show a perfect -35.

b. The -35 element, in general, is poorly conserved compared to the -10 element and TSS. This is because many promoters have either a poor -35 element, or no -35 at all, and instead rely on transcriptional activator proteins to recruit RNA polymerase to the DNA.

c. The σ factor makes base specific interactions with the -35 element at positions 1,2,3 and 5 (PMID: 11931761). Hence, the upstream half of the -35 element is much better conserved.

In summary, there is nothing unusual about the -35 analysis in our paper. However, to try and avoid confusion, we've updated Figure 1 so that the individual promoter components are aligned individually rather than the whole promoter being aligned based on TSS position. This makes the -35 element easier to see.

2. clarification of classification in cappable-seq:

It is unclear whether the cappable-seq approach filters out transcript reads without recognizable promoter elements.

Cappable-seq is completely unbiased and simply identifies triphosphorylated RNA 5' ends, regardless of the DNA sequence adjacent to the site where the RNA 5' end maps.

How are genic and intergenic reads defined by the authors?

Genic reads are those where the RNA 5' end (and so TSS) maps to the coding sequence of a gene. Intergenic reads are RNA 5' ends mapping between genes.

3. Interpretation of intergenic/nongenic transcription in context of post-termination complexes (PTCs):

We appreciate the reviewer's interest in PTCs and have considered the comments carefully. For the reasons outlined below, we don't think we infer much about PTCs with the data available. Even so, we've tried to make some changes to the text to acknowledge the reviewer comments where we can. We stress that PTCs are not the subject of our study and hope the reviewer can appreciate our reasoning.

Much of the cappable-seq signal appears intergenic.

This is partially correct, but the majority of TSSs are intragenic, consistent with observations for other bacteria.

There is a growing body of literature showing that RNAP can initiate transcription without sigma via post-termination complexes (PTCs) (PMIDs: 39779919, 37406096, 37459525). These initiate

at -10-like elements without -35 motifs and may recruit sigma and CarD (PMID: 31974358). The authors must consider:

We would revise this statement. In our view, there is evidence that post-termination complexes can be formed *in vitro* (PMID: 31974358) and if conditions are correct core RNA polymerase can subsequently initiate transcription (PMIDs: 31974358 & 39779919). The degree to which this might occur *in vivo* is unknown. However, available evidence strongly suggests cells take precautions to remove PTCs from the DNA and that a failure to do so is toxic (PMIDs: 39779919 & 37406096). As such, we think the consensus view of the field is that the vast majority, if not all, site specific transcription initiation *in vivo* is dependent on canonical promoter:RNAP interactions. As an aside, we also note that PMID: 39779919 reports PTC reinitiation “independent from... conserved promoter elements” rather than from -10 elements without -35 motifs. Hence, whether certain DNA sequences are associated with PTC reinitiation remains an open question.

-Whether some of the observed TSSs originate from PTCs rather than bona fide promoters.

This is an interesting question, but it is impossible to use our data to differentiate between bona fide promoters and any TSSs due to PTCs (further details below). As outlined above, it seems very unlikely PTCs give rise to significant transcription initiation *in vivo*. All the available evidence is that cells go to great lengths to prevent this. Indeed, this is discussed in the paper the reviewer cites (PMID: 39779919) that notes “evolved strategies to suppress promoter-independent transcription appear manifold”. We have added text explaining this at the end of the first results section.

-The presence of bidirectional symmetry and absence of -35 motifs as features of PTC-derived initiation.

We have done additional analyses and see no evidence for subclasses of TSSs (intragenic, intergenic, bidirectional, or otherwise) that consistently lack -35 elements. Of course, as mentioned above, the caveat is that individual promoters often lack a good match to the -35 element and rely either on activator proteins or alternative promoter elements (TG motifs, UP elements, AT-rich spacers) to recruit RNA polymerase. As such, the absence of a -35 is a common feature of canonical promoters and cannot be used as a simple marker of PTC derived initiation *in vivo*.

-That PTCs are highly sensitive to supercoiling (PMID: 39779919).

We agree that this is true, but PTCs are not the subject of our paper and we don't feel that we can add anything to the wider literature about PTCs using our data. Indeed, as noted above, PTCs are likely to be very transient inside cells as multiple mechanisms exist to remove them.

-That *R. sphaeroides* appears to encode a RapA-like factor (WP_119001171.1), which clears PTCs.

We agree that this is true and argue this makes it even more likely that TSSs due to PTCs are likely to be very rare, if detected at all. We now briefly mention mechanisms to remove PTCs from the DNA in the results section.

That CarD and sigma association with intergenic TSSs may reflect stabilization of PTC bubbles rather than standard promoter interactions.

We do appreciate this point, and do acknowledge this is theoretically possible. However, all the available evidence, and the current position of the field as a whole, is that the vast majority of chromosomal sites, associated with σ and CarD, are likely to be promoters. Consistent with this, we find strong conservation of promoter elements at these locations.

4. Missing -35 and extended -10 motif analysis:

Explicit analysis of these motifs would help determine which TSSs are likely sigma-dependent versus PTC-derived. Without it, many TSSs may be misclassified.

We have modified Figure 1f to specifically align the various promoter elements. Even so, we respectfully disagree with the reviewer's logic here. As outlined earlier in our response, it is not at all unusual to find promoters that lack a good -35 element. Indeed, this is very common because of the role played by transcriptional activators. For example, the promoter motif below is taken from our prior paper (PMID: 21398630) and the list of 554 *E. coli* promoters used to make the motif is described by PMID: 12907708. Importantly, this list pre-dates the genomic era and represents well-defined promoters for mRNAs. Compared to the -10 hexamer, the -35 is very poorly conserved. Hence, a missing -35 cannot be used to identify TSSs due to PTCs, which current data indicate must be short lived *in vivo*. Furthermore, there is no consensus view that a -10 element, in the absence of a -35, is a marker of PTC derived reinitiation as the reviewer suggests. For example, PMID: 39779919 reports PTC-like reinitiation by core RNAP without any promoter elements at all. Overall, we do not think it is possible to identify PTC-derived TSSs from our data and, even if it were, the expectation is that these would be very rare in wild type cells.

[REDACTED]

5. Mechanistic interpretation of CarD and supercoiling (including Fig. 4C):

Recent findings in *M. tuberculosis* suggest that CarD's effects are inhibited on negatively supercoiled DNA (PMID: 40060575). The authors should:

We would describe the pre-print differently. Rather than showing "CarD's effects are inhibited on negatively supercoiled DNA" the pre-print provides an example of a promoter that switches from being activated to repressed by CarD when DNA is negatively supercoiled. As expanded on

below, this was also known from prior work and fits perfectly with our model. We covered the original paper (PMID: 37075846), reporting a similar result to PMID: 40060575, in our Discussion. We now also cite the newer work.

-Discuss this preprint in more detail and reconcile the discrepancy with their findings in a more mechanistic way.

We now cite the pre-print (since published in JBC) but do not think there is a discrepancy with our findings. Indeed, we had already explained how this exact phenomenon (reported by PMID: 37075846, cited in the original text) fits within the framework of our model. This is in the Discussion section and the updated text is pasted below for clarity.

-Clarify that novobiocin reduces negative supercoiling, likely destabilizing RPo indirectly, thus complicating their conclusions.

We probably don't describe this clearly enough. Briefly, we don't think this complicates our conclusions, rather this is exactly what we propose. To expand, CarD activates transcription by stabilising otherwise unstable open complexes. Negative DNA supercoiling does the same, albeit by a different mechanism. Hence the co-dependence of many promoters on negative DNA supercoiling and CarD. As the reviewer correctly notes, novobiocin reduces negative supercoiling, and so RPo stability. As CarD is part of the open complex formed by RNAP and the DNA, if this becomes less stable, the CarD binding signal will decrease. We now reiterate this model throughout the text.

-Consider the potential impact of novobiocin on PTC activity.

We do appreciate the reviewer's interest in PTCs but, as outlined above, genuinely don't feel we can comment on this issue using the data we have.

-Quantify transcription in all lanes in Fig. 4C to compare CarD+ vs. CarD- at increasing negative supercoiling levels.

This has been done and the revised figure shows fold activation by CarD (i.e. the CarD+/CarD- signal is plotted).

-Discuss why RNAI is used as a control and provide its promoter sequence—CarD has no apparent effect here, which needs explanation.

RNAI is widely used as a control for *in vitro* transcription assays and has been for many decades (e.g. PMIDs: 15126488, 9218807 & 10542171, but there must be 100s of papers). The RNAI is derived from the colE1 plasmid replication origin (where its biological function is to control plasmid copy number). Plasmids used as templates for *in vitro* transcription assays invariably contain colE1. As promoter driving RNAI production is constitutive, RNAI has become a ubiquitous control.

We note that others also found no impact of CarD on RNAI transcription (PMID: 33168725) so this is not unusual. Presumably, as the promoter is constitutive, there is simply no dependence on CarD. It's also worth noting the RNAI promoter is not native to the Rhodobacter system. The promoter has the sequence 5'-tc**TTGAAG**tggtggcctaactacggc**TACTACT**agaaggAca-3'

We haven't altered the text, since we had already described RNAI has being derived from the plasmid replication origin, but hope this provides sufficient further explanation for the reviewer.

-More clearly quantify results from Figs. 4 and 5 to support conclusions about the effects of -10 element mutations.

The quantification has been added. Although these figures are now largely new data.

6. Analysis and rigor (Figs. 4 and 5):

-Indicate the number of replicates for each experiment.

There are three replicates each for Figures 4 and 5. This detail has been added.

-Quantify transcription in all lanes in Fig. 4C to compare CarD+ vs. CarD- at increasing negative supercoiling levels. The current quantification in 4c is just CarD. Also where is the error bar in the red bar?

The reviewer also requested this change earlier in the review, and we have made the alteration, but please see our prior response for the details. There was no error shown for the red bar because this lane was set to 100 % for each replicate, allowing for normalisation between experiments. An error bar is now included because we show activity +/- CarD, rather than % activity, in the revision.

-Provide P-values and statistical significance.

Added where appropriate.

-In the supplement, label each gel lane and replicate clearly (e.g., replicate 1, 2, 3).

This has been done.

7. Mechanisms (Fig. 3D and Line 143):

-The statement that CarD depletion causes "open complex collapse" should also consider that it might promote faster escape into elongation.

We agree that this could be the case for promoters repressed by CarD, where the open complex is over stabilised. However, most promoters we detect are activated by CarD. In these cases, loss of CarD results in less transcription, which isn't consistent with a faster rate of promoter escape. Note that, for Figure 3D, faster escape into elongation does not fit the data, which shows decreased transcription. Additionally, the section has been rewritten and no longer mentions open complex collapse.

-Novobiocin treatment results in both upregulation and downregulation of transcription—even at CarD-associated promoters—this duality needs discussion. It is currently ignored.

The new Figures 3e, 3f, 5 and 6 now deal with this topic. We compare the two classes of promoter both *in vivo*, using genome scale data, and *in vitro* using biochemical assays.

- Different promoters may respond to CarD in different ways depending on their kinetic properties (PMID: 37075846); this should be acknowledged.

We have added this point to the discussion.

8. Discussion:

The discussion should incorporate:

-The broader literature on CarD function, supercoiling effects, initiation kinetics, and promoter escape.

We have added extra details, partly in response to a previous question that the reviewer raised. We also compare CarD activated and repressed promoters, as well as the different impact of CarD in *M. tuberculosis*, and *T. thermophilus*.

-The likely contribution of topoisomerases and DNA topology to transcription across bacteria.

We did already touch on this (e.g. supercoiling control of rRNA transcription in *B. subtilis*, and supercoiling changes in starved *E. coli*).

-A critical evaluation of whether all observed TSSs represent bona fide promoters.

We've probably answered this question in our responses above; we think the vast majority of, if not all, observed TSSs represent bona fide promoters. Of course, there is also the issue of what a "bona fide promoter" really is. Do these have to be promoters in intergenic regions? Do these include DNA sequences that resemble a promoter but don't generate a useful mRNA? Does there have to be a certain level of activity for a promoter to be considered real? These are quite broad philosophical questions that we have discussed at length in prior work. However, we are not sure there is much to add in the current paper; there isn't really a logical place to introduce this sort of ideas. However, should the paper be accepted, our detailed discussions above will also be available online for interested parties.

-The roles of DksA seem less relevant and can be de-emphasized in favor of the points above.

Only two very short sentences mention DksA, and we think it's important to compare strategies for coupling rRNA transcription to growth phase in different bacteria, so we would like to keep this detail in if possible (we are not restricted for space).

Reviewer #3 (Remarks to the Author):

Forrest et al is a well-designed study on the effects of the transcription factor CarD in regulating *Rhodobacter sphaeroides* promoters. The authors combine various high-throughput data, including ChIP-seq, RNA-seq, transcript start site mapping (cappable-seq), and negative supercoiling mapping (with psoralen sequencing) to define the CarD regulon and supercoiling-sensitive genes of *R. sphaeroides*. These authors then use biochemical experiments with plasmids of different negative supercoiling to demonstrate that the *rrnBP1* promoter requires both CarD and negative supercoiling for activation, with this requirement bypassed by a -7A → T transversion in the -10 sequence. The authors thus conclude that "an optimal -10 element sequence, negative supercoiling, and CarD all enhance promoter activity, and that maximal *R. sphaeroides* transcription requires only two of these stimulatory factors."

I found the global study technically sound and the biochemical experiment especially compelling. Although a relationship between CarD, supercoiling, and open complex formation has been reported, the high-throughput studies are novel in *R. sphaeroides* and the biochemical experiment clearly demonstrates the relationship between CarD and negative supercoiling at the *rrnBP1* promoter. However, these studies are not sufficiently complementary, with the conclusions are not justified by the experiments. I believe that the authors are conflating multiple distinct classes of CarD-dependent promoters in this study. I also have many

questions about the proposed model and experimental approaches. Either more experiments or significantly more limited interpretations of the data is warranted.

We agree with the reviewer. There are different classes of CarD regulated promoter and these could have been considered. To explain our initial reasoning, in the case of *R. sphaeroides*, most (~75%) CarD-associated promoters, detected *in vivo*, are relaxation repressed. Hence, we focused the original manuscript on activation due to interplay between CarD and negative supercoiling. We chose *rrnBP1* for the *in vitro* work because of the promoter's importance for cell physiology, as it had been used in the past to study CarD in *R. sphaeroides*, and since it is an example of a relaxation repressed promoter (discussed further below). We appreciate that we could have stated the intention of our focus more clearly. As requested, the new version of the paper includes different promoter classes.

Major concerns:

1) As noted by the authors, CarD-dependent promoters can be repressive or stimulatory, but no effort is made to distinguish between these classes. Indeed, previous work in Mycobacteria suggests that CarD-mediated open complex formation could either promote promoter escape (activation) or inhibit it (repression) (Zhu et al, PMID: 31217290), suggesting that CarD/supercoiling could identically impact open complex formation at many promoters but cause vastly different transcriptional outcomes.

In the revised paper we have extended the scope of our *in vitro* analysis to include an additional relaxation repressed promoter (*rpsL*), a relaxation activated promoter (*carD*), and a series of chimeric promoters derived from *rrnBP1* and the *carD* promoter (Figures 5, 6 and S6). We have also investigated both *rrnBP1* and the *carD* promoter using mutagenesis (Figures 5 and 6). Furthermore, the genome-scale data have been subject to further analyses as suggested (Figures 3f and 3g). We feel the altered text better links the different parts of the story.

The details of the additional analyses and experiments are laid out in response to the various questions below.

a. Only the *rrnBP1* promoter is tested biochemically and it is not made clear the relationship between this promoter and the high-throughput experiments. To what extent is *rrnBP1* an appropriate model for the relaxation-repressed, CarD-bound promoter class proposed in this study? Is the *rrnBP1* promoter bound by CarD in ChIP-seq and significantly downregulated in response to chromosome relaxation?

We have now made the relationship to the genomic data clear. Briefly, in our high-throughput experiments, binding of σ^{93} and CarD at *rrnBP1* is indeed downregulated by novobiocin, consistent with this being an appropriate model. We can't specifically comment on precise transcript levels, since RNA-seq experiments i) require rRNA depletion and ii) any reads detected are difficult to map because of the multiple rRNA operons. We also note the link to the prior study of CarD and *rrnBP1*, which informed our decision to use this promoter.

What is the sequence of the *rrnBP1* promoter and how does it relate to the predicted CarD-dependent promoters (see further comments regarding sequence logo in 1b, below)? Biochemically validating additional CarD-associated novobiocin-repressed promoters is essential to strengthen the conclusions presented here.

The sequence of *rrnBP1* is now shown in the new Figure 5a. As requested in comments below, we have made new sequence logos (from the genome-scale data) for sites bound and unbound

by CarD (Figure 1f). The frequent lack of position -7T is clear for the CarD bound promoters and this is consistent with the sequence of *rrnBP1*.

As suggested, we have also biochemically validated an additional CarD-activated novobiocin-repressed promoter, for the gene *rpsL*. The data resemble those obtained for *rrnBP1* (see new Figure S6). We have also tested a series of chimeric *rrnBP1* derivatives containing sections of *PcarD* (Figure 6). The chimeric promoters remain CarD activated (Figure 6), although sequence changes can reduce the stimulatory impact of supercoiling.

It is also clear from this study and the literature that there is a class of CarD-repressed promoters, with the prime example the CarD promoter itself (Henry et al, PMID: 34152199). Henry et al showed that an extended -10 element and other spacer sequences mediated repression. In this work, the authors have a set of CarD-dependent relaxation-activated promoters (Figure 3) that are overlooked.

We agree, and the responses below detail additional analysis of the genome-scale data (Figures 3f, 3g and S4) and *in vitro* experiments with the *carD* promoter (Figures 5 and 6). These include and comparisons with *rrnBP1*.

Do these promoters have the same conserved sequences as those that are relaxation-repressed?

We have done additional analysis of the genome-scale data (Figures 3f, 3g and S4) to answer this question. Relaxation-activated and -repressed promoters, bound by CarD, are extremely similar (Figure 3f). Overall, the only significant (but still very small) difference is that relaxation activated promoters tend to have better -10 hexamers (Figure 3g). We conclude that there is no simple “hallmark” that defines one promoter class compared to another. This is further evidenced by comparing *PcarD* and *rrnBP1* (Figures 5 and 6) *in vitro*. Hence, introducing the -7T mutation, a TG motif, and a more AT-rich discriminator (the latter two being features of *PcarD*) fail to convert *rrnBP1* into a CarD repressed promoter, though the requirement for both CarD and negative supercoiling is removed.

Is the CarD promoter supercoiling-dependent and is it repressed or activated?

We have now done these experiments (Figures 5, 6 and S6). Consistent with Henry *et al.*, we find that the *carD* promoter is repressed by CarD. We show that this repression is very sensitive to supercoiling. In fact, on a relaxed DNA template, the role of CarD switches to that of a *PcarD* activator (Figure 5). This is consistent with the overall model that i) negative supercoiling and CarD both enhance DNA opening and ii) at some promoters CarD and negative supercoiling over stabilise DNA opening and the result is repression.

Are the sequence features identified in Henry et al observed in any promoter classes? An ideal experiment would be to deplete CarD and reexamine transcription to distinguish between these promoter sets but some effort should be made to categorize and validate that the promoter features associated with different CarD/supercoiling regulation classes.

As noted above, we have compared promoter classes and there are no consistent sequence differences. Rather, CarD-repressed relaxation-activated promoters tend to have a slightly better match to the -10 element consensus (Figure 1f and 1g). We've also looked carefully at the Henry *et al* (PMID: 34152199) data. Consistent with our conclusion, the *carD* promoter is a better match to the promoter consensus than *rrnBP1* (Figure 1a of Henry *et al*). Of even more interest is Henry *et al* Figure 4. As the reviewer notes above, the authors make a series of

promoter mutations around the *carD* promoter -10 element and spacer region. None of these mutations allow CarD to activate instead of represses. However, the mutations do influence overall *carD* promoter activity and the degree of repression by CarD. Comparison of Figures 4c and 4d in Henry *et al.* is particularly striking, as there is a near perfect inverse correlation between overall promoter activity (Figure 4c) and fold-repression by CarD (Figure 4d). This is true despite the different mutations impacting different parts of the promoter. Overall, this seems fully consistent with our conclusion above (CarD repressed promoters tend to be better promoters, but no single feature identifies a CarD repressed promoter). We have also made chimeric promoters with parts of *PcarD* replacing the equivalent *rrnBP1* region (Figure 6). Whilst these changes can increase *rrnBP1* activity, and reduce the fold activation by CarD, a promoter subject to repression by CarD is not created.

The authors should also consider biochemically testing CarD-associated novobiocin-induced promoters. In these cases, is CarD functioning in synergy or opposition to supercoiling? Without these experiments, the authors should be extremely cautious in generalizing their biochemical result to encompass the full range of possible CarD/supercoiling interactions.

As also noted above, we have done this for *PcarD*. Repression by CarD is synergistic with supercoiling as expected (Figure 5). Indeed, when the *PcarD* DNA is relaxed, CarD acts as an activator rather than a repressor. This is consistent with our overall model that both negative supercoiling, and CarD, stimulate DNA opening. Usually, this leads to activation. However, at some promoters (on average having a better -10) repression occurs because DNA opening is presumably too stable.

b. The authors make an argument for -7T/-12T de-enrichment in CarD promoters. However, there are several issues with idea. In figure 1F, I cannot see the difference in the -7T/-12T as reported. The de-enrichment of -7T is well-supported by the literature (Henry *et al*, PMID: 33168725; 34152199) but T is still an enriched base in the sequence logo presented. The -12 T is clearly enriched as well. Is the interpretation based on p-value? It does not make sense to quantitatively compare two p-values. I would suggest the bigger differences are that CarD-bound promoters lack -8T and a G at -33. Perhaps a quantitative analysis like in Henry *et al* (PMID: 33168725) would be more appropriate.

We agree with the reviewer, comparing motifs based on p-values could be misleading. We have redesigned the analysis to better identify individual promoter motifs (these are now aligned individually, rather than the entire promoter region being aligned based on TSS position). It is clear in the revised figure panel that CarD bound promoters are de-enriched for -7T. In the new sequence logos, based on the frequency at which different bases occur, rather than p-value, -12T is similarly prevalent in CarD bound and unbound promoters. Even so, we have still investigated the role of position -12T further, since it also has an important, but different, role in DNA opening. Whilst CarD can compensate for a missing -7T, the factor cannot correct promoter defects if -12T is mutated (Figure 5).

More worryingly, nothing is ever explored for -12T (or for any other promoter differences). Henry *et al* (PMID: 33168725) observed that -12T (without -7T) was conserved for 15 promoters activated by CarD *in vitro*. Further, the CarD promoter itself contains a -12T. What is the explanation for these discrepancies? What is the function, if any for these sequences? How do they relate to the different potential CarD/supercoiling promoter classes (1a, above)? Does -12T have any effect on CarD/negative supercoiling synergy?

As noted above, when assessing frequency of occurrence rather than p-value, -12T is similarly prevalent in CarB bound and unbound promoters. Even so, for completeness, we have examined the impact of mutating this position at both the *carD* promoter and *rrnBP1* (Figure 5).

At the end of the work, I am convinced that for *rrnBP1* both CarD and negative supercoiling are required for activation. However, I have many remaining questions about the overall model.

2) Why is *rrnBP1* be regulated in this fashion? I agree with the authors that synergy is logical, but this operates more like an “AND” logic gate, meaning the effects are not tunable. Is it to quickly shutoff ribosomal synthesis during unfavorable conditions? I feel this idea would be better-supported if there are additional examples of this regulation. Perhaps this lack of synergy is unique to *rrnBP1*.

We’re not sure we agree fully on this point, but we see why the reviewer asks this question; it’s perhaps a consequence of the way the result is presented. We think the data do show tunability. For example, Figures 4b and 4c show *rrnBP1* activity across a range of negative supercoiling levels in the presence and absence of CarD. As negative supercoiling gradually increases, more transcript is progressively produced. The promoter responds to CarD levels in a similar way (e.g. Figure 2a of PMID: 33168725). Of course, drastic changes in supercoiling, that may be encountered in certain stress conditions, would allow the promoter to be quickly shut off, as the reviewer notes.

3) How generalizable is this finding for all *R. sphaeroides* promoters? I also agree that “the combined impact of CarD and DNA topology must differ according to biological context”. Examining several more promoters and, ideally, in different regulatory classes (novo-activated, novo-inhibited, CarD-bound, CarD-free, etc) is essential for extending the experiments to infer CarD’s effects on individual promoters. This is especially true since CarD is itself controlled by a negative feedback loop. Whether negative supercoiling at the CarD promoter increases or represses CarD expression will have consequences for the interpretation of the transcriptomic/ChIP data, especially if CarD is rapidly turned over.

As suggested earlier in the review, we have now explored the impact of negative supercoiling at an additional relaxation-repressed CarD-activated promoter (*PrpsL*) and at a relaxation-activated CarD-repressed promoter (*PcarD*). In all cases, the promoters are supercoiling sensitive, in the expected way (Figures 5 and S6).

We have also constructed and tested chimeric promoters based on *rrnBP1* (see the new Figure 6). In these experiments, we have swapped parts of the CarD-repressed relaxation-activated *PcarD* promoter for the equivalent region in *rrnBP1*. As eluded to above, there is no hallmark sequence change that can convert *rrnBP1* into a *carD* repressed promoter. However, introducing the various *PcarD* sequences gradually improve the overall activity of *rrnBP1* and negates the need for CarD. Importantly, although the changes alter the combinatorial impacts of CarD and supercoiling, these are never removed. Hence, we think it’s reasonable to conclude that most CarD bound promoters are supercoiling sensitive.

With respect to the genome-scale experiments, and as expected, novobiocin treatment does indeed lead to increased *carD* transcription. In the same conditions, transcription from most CarD associated promoters decreases, consistent with our overall conclusion that CarD is primarily an activator during normal growth. Indeed, the impact of supercoiling must be acute if CarD activated promoters are downregulated, despite cells producing more CarD.

4) How does CarD/RNAP together sense negative supercoiling? Figures 2-3 show that CarD-bound sites are more negatively supercoiled than random sites. However, are these CarD-bound sites more negatively supercoiled than σ^{93} -bound sites?

We have now done this analysis and the answer is yes, sites bound by CarD and σ^{93} are significantly more negatively supercoiled than those bound by σ^{93} alone. See the new Figure S3.

One would imagine that any active TSS must be negatively supercoiled since negative supercoiling promotes promoter melting and RNAP escape.

We agree, although suggest adjacent transcription units are also likely to influence this. For example, a promoter between tandem genes could be impacted by positive supercoiling from the upstream transcription unit. Overall, σ^{93} bound regions, whether associated with CarD or not, are more negative supercoiled than random genomics regions.

If CarD-free σ^{93} -bound sites are more negatively supercoiled, CarD could perhaps compensate for the absence of negative supercoiling at promoters rather than require it for binding, which is the model presented here.

We did not intend to imply that negative supercoiling is required for CarD binding (at least not in the sense to which we think the reviewer refers). Rather, we think open complex stability (and so stability of the entire DNA-RNAP-CarD ternary complex) is sensitive to supercoiling. Hence, when negative supercoiling decreases, it is not only the binding of CarD that decreases but also σ^{93} . We suspect we did not make this point clearly enough, as reviewer 1 made a similar comment. The text has been altered in several places.

This analysis should also be performed again in the novobiocin-treated dataset, as it would be interesting to know if σ^{93} /RNAP remains more bound when CarD is present/absent, which would support the hypothesis that CarD stabilizes open complexes.

As noted above, we probably weren't clear enough on this issue, when CarD binding decreases so does binding of σ^{93} (e.g. see Figure 3e).

Relatedly, for Figure 6B, because CarD is associated with σ^{93} (housekeeping) promoters, this set should be naturally enriched in metabolism and central dogma processes. I'd be curious to see the comparison to the remaining σ^{93} -dependent non-CarD promoters, which are also likely enriched for these functions. Significant differences between these two sets would be more compelling than the comparison to random promoters, which are likely enriched for alternative sigma factor promoters that regulate stress response genes.

This is a really good point, and we have done the suggested analysis (see the new Figure 7). When compared to promoters bound by σ^{93} alone, promoters also bound by CarD are enriched for adjacent gene functions related to central dogma processes. In particular, promoters for genes encoding tRNAs and ribosomal proteins. We appreciate this comment as it allows us to make a more nuanced conclusion.

5) Based on the Methods, it is not clear how the σ^{93} and CarD overlapping promoters were selected. Further definition of what "overlap" constitutes would be helpful.

We have updated the methods to make this clearer.

I also wonder where CarD binds at these promoters (e.g. around TSS, -10, or hundreds of bases upstream) and if this information could be used to refine the models for CarD/supercoiling behaviors.

The spatial resolution of ChIP-seq, related to the length of sonicated DNA fragments (a few hundred base pairs) is insufficient to distinguish between CarD binding at TSSs compared to -10 elements. Even so, we have quantified the distances between CarD binding peaks and TSSs. The majority are within 100 bp of each other. I.e. the locations are essentially indistinguishable.

6) Is there a relationship between CarD and the bidirectional promoters noted in Figure 1? Is the hypothesis that CarD should activate these promoters and it should activate both? Is CarD bind to these bidirectional promoters?

Bidirectional promoters are slightly more likely to be CarD associated (see new Figure S2d) and we would predict activation by CarD in both orientations.

7) Line 147: Is the idea here that more CarD-bound genes are downregulated than CarD-independent? Can this be explained more clearly (e.g. more CarD-bound genes were downregulated)? There also appears to be more genes upregulated for CarD-independent promoters, is this correct?

This part of the paper has been substantially reworked. Briefly, a greater percentage of CarD bound promoters were downregulated, but the absolute number is higher for CarD free promoters (as there are more of them). We have revised the figure to make space for the comparison of relaxation-repressed and relaxation-activated promoters. In doing so we have taken out the volcano plots for CarD-independent promoters, since there is potential for confusion.

8) Line 151-154: I did not understand the data for “Novobiocin predominantly downregulates transcription and this is more pronounced for CarD bound genes”. Is the idea that more CarD-bound genes are downregulated upon treatment? To my eye, it appears that there are more genes downregulated for CarD-free promoters. I also suggest the statement “novobiocin caused significant DNA relaxation, loss of both CarD and σ^{93} binding, and lower gene expression” should be revised. Novobiocin-induced/DNA relaxation-induced genes have been observed in diverse organisms, and there are many genes (red) that are induced upon DNA relaxation in Figure 3D.

The answer here is similar to the above; a greater percentage of CarD bound promoters were downregulated, but the absolute number is higher for CarD free promoters. As noted in response to the prior question, this part of the paper has been reworked.

Minor comments:

1) The authors should carefully check over their referencing and reference the primary literature as often as possible:

a. Line 27-28: a review here would be more appropriate, as reference 2 (Zhang et al) were only able to visualize domains 2-4 and do not discuss the role of sigma domains in depth. Reference 3 also does not the primary source to show sigma and -10/-35 contacts.

b. Line 30: Instead of references 2-4, which are not the references that show sigma domain 2 interacting with -7T/-11A, please cite Feklistov and Darst (PMID: 22136875), Zhang et al (PMID: 23086998), and Zuo and Steitz (PMID: 25866247).

We have re-referenced the section related to the above two points.

c. Line 108: The authors may also want to cite Lal et al (PMID: 27025941), the first report of psoralen use genome-wide in bacteria.

We now cite this paper.

2) Line 46-47: The authors are correct that lack of rotation of RNAP increases supercoiling. However, simple duplex DNA melting by RNAP also leads to torsional strain and consequent twisting of DNA.

We have added the word “primarily” to ensure we don’t exclude other processes.

3) Line 75: Is there some way to distinguish between “active” and “infrequently used” promoters from cappable-seq? Perhaps the studies would provide better motif resolution if the highest-active promoters are selected.

This is perhaps not as straightforward of a distinction as it may seem. For instance, there will be many intrinsically poor promoters that give strong signals (i.e. are very active) because activator proteins intervene. Similarly, many intrinsically strong promoters will be infrequently used because of repressors. It’s also not unusual for promoters, occurring by chance, inside genes, to be more active than the canonical promoter upstream of the gene. The adjacent motifs are derived from the top or bottom 10% of promoters, according to cappable-seq signal strength. The motif differences are surprisingly modest, given the differences in promoter activity. This is particularly true for promoters that occur inside genes.

4) Please renumber the logo (indicating +1) in Figure 1F and mark the correct bases to examine to ease the burden on the reader.

The figure panel has been redesigned to also include this suggestion.

We than the reviewers again for their supportive comments. Our response to the final concern of reviewer 2 is provided below.

Reviewer #1 (Remarks to the Author):

The authors have address my concerns, no further comment. Thank you.

Reviewer #2 (Remarks to the Author):

The authors have satisfactorily addressed my previous comments. My only remaining concern is the use of the word “flaw” throughout the manuscript (including the Abstract and Introduction) to describe the lack of consensus among promoter elements. I recommend replacing this term with more precise language—for example, explicitly stating that certain promoters deviate from the canonical -10 consensus or lack specific consensus features such as the $-7T$. The term “flaw” implies a subjective judgment, whereas a more descriptive formulation would be more precise.

To explain our rational, particularly for the abstract, we want to get to the key point quickly. Hence, we only say that CarD aids -10 opening and, in *R. sphaeroides*, this is particularly important because there is an inherent -10 element defect. If we were to get more specific, and describe the defect, we must also explain that different -10 element bases have different roles, introduce the importance $-7T$, and discuss base flipping. This is probably too much for the abstract, which is already on the longer side. In the main body of the text, we agree, and were mostly careful, every time this was mentioned, to specify that the defect is missing $-7T$ base. However, we found one incidence, on page 3, where we hadn't done this and have added extra text.

Reviewer #3 (Remarks to the Author):

I commend the authors on the extensive and thorough revision. All the new experiments and analyses greatly improve the manuscript, and I have no further comments. Congratulations on the fine work.

This study offers compelling observations on the roles of CarD and DNA supercoiling in transcriptional regulation in *R. sphaeroides*. However, several aspects of data interpretation require closer alignment with existing mechanistic and structural work. In particular, a more thorough analysis of promoter architecture—including the -35 and extended -10 motifs—alongside consideration of non-canonical initiation mechanisms such as post-termination complexes (PTCs), is essential. The integration of ChIP-seq, Psoralen-seq, RNA-seq, and cappable-seq represents a significant strength, enabling a genome-wide view of regulatory dynamics. Together, these approaches provide complementary insights into promoter activity, DNA topology, and protein-DNA interactions. With the revisions and clarifications outlined above, this work could make a substantial contribution to our understanding of transcriptional regulation in bacteria. This work presents exciting observations but the interpretations need to be grounded more firmly in the mechanistic literature. A more rigorous analysis of promoter architecture (especially the -35 and extended -10 motifs), consideration of non-canonical initiation via PTCs, and improved data presentation will significantly strengthen the manuscript.

Major Points for Revision:

1. **-35 element and promoter assignment (fig. 1):**

The lack of -35 element analysis is a puzzling. Why is this not included in the assignment of bona fide promoters? *R. sphaeroides* appears to have canonical -35 elements and lacks RbpA (PMID: 33168725), so their inclusion in promoter analysis is essential.

2. **clarification of classification in cappable-seq:**

It is unclear whether the cappable-seq approach filters out transcript reads without recognizable promoter elements. How are genic and intergenic reads defined by the authors?

3. **Interpretation of intergenic/nongenic transcription in context of post-termination complexes (PTCs):**

Much of the cappable-seq signal appears intergenic. There is a growing body of literature showing that RNAP can initiate transcription without sigma via post-termination complexes (PTCs) (PMIDs: 39779919, 37406096, 37459525). These initiate at -10-like elements without -35 motifs and may recruit sigma and CarD (PMID: 31974358). The authors must consider:

-Whether some of the observed TSSs originate from PTCs rather than bona fide promoters.

-The presence of bidirectional symmetry and absence of -35 motifs as features of PTC-derived initiation.

-That PTCs are highly sensitive to supercoiling (PMID: 39779919).

-That *R. sphaeroides* appears to encode a RapA-like factor (WP_119001171.1), which clears PTCs.

That CarD and sigma association with intergenic TSSs may reflect stabilization of PTC bubbles rather than standard promoter interactions.

4. **Missing -35 and extended -10 motif analysis:**

Explicit analysis of these motifs would help determine which TSSs are likely sigma-dependent versus PTC-derived. Without it, many TSSs may be misclassified.

5. **Mechanistic interpretation of CarD and supercoiling (including Fig. 4C):**

Recent findings in *M. tuberculosis* suggest that CarD's effects are *inhibited* on negatively supercoiled DNA (PMID: 40060575). The authors should:

-Discuss this preprint in more detail and reconcile the discrepancy with their findings in a more mechanistic way.

-Clarify that novobiocin reduces negative supercoiling, likely destabilizing RPo indirectly, thus complicating their conclusions.

-Consider the potential impact of novobiocin on PTC activity.

-Quantify transcription in all lanes in Fig. 4C to compare CarD⁺ vs. CarD⁻ at increasing negative supercoiling levels.

-Discuss why RNAI is used as a control and provide its promoter sequence—CarD has no apparent effect here, which needs explanation.

-More clearly quantify results from Figs. 4 and 5 to support conclusions about the effects of -10 element mutations.

6. Analysis and rigor (Figs. 4 and 5):

-Indicate the number of replicates for each experiment.

-Quantify transcription in all lanes in Fig. 4C to compare CarD⁺ vs. CarD⁻ at increasing negative supercoiling levels. The current quantification in 4c is just CarD. Also where is the error bar in the red bar?

-Provide P-values and statistical significance.

-In the supplement, label each gel lane and replicate clearly (e.g., replicate 1, 2, 3).

7. Mechanisms (Fig. 3D and Line 143):

-The statement that CarD depletion causes "open complex collapse" should also consider that it might promote faster escape into elongation.

-Novobiocin treatment results in both upregulation and downregulation of transcription—even at CarD-associated promoters—this duality needs discussion. It is currently ignored.

- Different promoters may respond to CarD in different ways depending on their kinetic properties (PMID: 37075846); this should be acknowledged.

8. Discussion:

The discussion should incorporate:

-The broader literature on CarD function, supercoiling effects, initiation kinetics, and promoter escape.

-The likely contribution of topoisomerases and DNA topology to transcription across bacteria.

-A critical evaluation of whether all observed TSSs represent bona fide promoters.

-The roles of DksA seem less relevant and can be de-emphasized in favor of the points above.